# ModalMix: Optimizing Multimodal Data Mixture with Compute-Dependent Regression

## Abstract

Training large multimodal models requires optimizing the data mixture to balance cross-modal synergies against finite computational resources. However, existing heuristics for data mixing largely ignore the underlying cross-modal dynamics and their dependence on compute scaling. In this work, we propose ModalMix, a framework that formalizes data mixture optimization by simultaneously modeling cross-modal interactions and compute-dependent scaling laws. ModalMix yields a predictive regressor for the optimal data mixture at any given computational budget. Empirically, models trained with ModalMix achieve 1.4× faster convergence than those with a uniform data distribution, alongside a 47% better average rank over 17 downstream tasks. The framework reveals that the optimal strategy is dynamic, not static: it initially prioritizes speech, then gradually shifts focus towards image-text data as compute increases, while maintaining stable use of text data. ModalMix offers a flexible and principled solution to the data-mixing problem, bridging a critical gap between scaling theory and practical multimodal pretraining.

## 1 Introduction

In the era of large-scale pretraining (Floridi & Chiriatti, 2020; Taylor et al., 2022; Li et al., 2023; Grattafiori et al., 2024), heterogeneous multimodal datasets spanning text, images, and speech are essential for generalized intelligence. However, effective training hinges on principled data mixture design, particularly in calibrating the relative contributions of each modality. While data mixture optimization is well established in unimodal settings (e.g., text-only), such methods are insufficient for multimodal learning. Cross-modal interactions, whether synergistic, conflicting, or independent, fundamentally shape how data should be allocated across modalities. These interactions are critical for balanced performance across diverse tasks (Li et al., 2022; Mu et al., 2023; Tao et al., 2024; Xu et al., 2025), but existing frameworks often fail to account for these complexities.

The challenge is further pronounced by computational constraints, which prior work often overlooks (Liu et al., 2025; Guo et al., 2025). In particular, the influence of computational budgets on the optimal modality mixture remains largely unexamined. As models scale in size and compute (Kaplan et al., 2020; Henighan et al., 2020) the ideal modality mixture shifts dynamically. These dynamics are largely underexplored and constitute a significant gap in the literature of multimodal pretraining.

To solve this problem, existing methods often rely on grid search or rule-based heuristics, which are costly and generalize poorly across scales (Guo et al., 2025; Xie et al., 2023b). Proxy model approaches (Liu et al., 2025) predict the optimal mixture based on certain performance metrics, though more efficient, fail to capture complex cross-modal interactions and largely ignore compute scaling effects arising from model size and training data. Furthermore, previous works often assume scale invariance (Ye et al., 2024), i.e., the optimal mixture remains static despite changes in model size or compute resources. In contrast, as recent works on unimodal vision or language models show that the optimal data mixtures are inherently scale-dependent, with optimal allocations shifting as models and compute grow (Sorscher et al., 2022; Kang et al., 2024). As shown in Figure 1, the performance of one modality is strongly influenced by the data mixture of others, even when its own data share is fixed, underscoring complex inter-modal dependencies. Motivated by these challenges, we pose the following research question: *Can we design an optimization framework that not only quantitatively captures cross-modal interactions, but also adapts to varying computational budgets, thereby accelerating training and improving performance across modalities?*

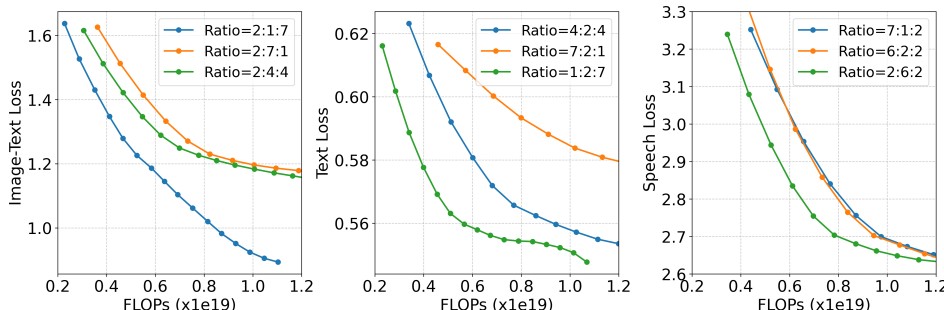

Figure 1: Training loss for different modalities over the training process of a omni-modal with a 1.5B language backbone. "Ratio" denotes the proportion of image-text, text, and speech data, respectively. For each studied modality, its ratio is kept at 20%, and the ratios of the other two modalities are varied. For each modality, changes in other modalities' ratios significantly affect its loss, indicating non-negligible cross-modality interactions during training.

To address this, we propose ModalMix, a regression-based framework for optimizing multimodal data mixture. Unlike heuristic approaches, ModalMix explicitly models the interplay between cross-modal dynamics, model scale, and computational budget. Its core is a predictive model trained on a comprehensive dataset of more than 1,620 experiments with varying data mixture, model sizes, and number of samples. ModalMix first predicts a "loss floor" for each modality, i.e., the minimum achievable loss under a given compute constraint. It then optimizes the mixture to minimize the aggregated loss, while preventing any modality from underperforming. This compute-aware strategy yields substantial gains: models trained with ModalMix achieve up to 1.4× faster convergence and demonstrate significant performance improvements in various downstream tasks.

More importantly, ModalMix reveals that the optimal data allocation is dynamic, not static. It prescribes a high initial proportion of speech data that declines as training progresses, a steadily increasing share of image-text data, and a stable proportion of text data to preserve linguistic grounding. These insights show that ModalMix offers not just an acceleration technique, but a principled, resource-aware strategy for scheduling complex multimodal data mixture. We summarize our contributions as follows: (i) We propose a novel multimodal data mixture optimization framework that leverages regression-based prediction to estimate the optimal data mixture across text, image, and speech modalities, incorporating both cross-modal dynamics and compute scaling; (ii) We introduce an automated balancing mechanism that dynamically adjusts the data mixture in response to changing computational budgets, achieving 1.4× faster convergence and performance improvements in 17 downstream tasks; and (iii) We provide new insights into compute-driven evolution of data mixture, highlighting dynamic shifts in optimal image-text, text, and speech allocations, offering practical guidelines for resource-aware data scheduling in multimodal systems.

## 2 RELATED WORK

**Data Mixture Curation.** Data mixture, which determines the proportion of data, is crucial for LLM pre-training as it strongly impacts downstream performance (Touvron et al., 2023; Xie et al., 2023a). While early work focuses on data filtering (Gao et al., 2020), recent research has shifted to strategic mixture optimization. Heuristic methods rely on computationally expensive grid searches or expert intuition (Raffel et al., 2023). In contrast, quantitative methods offer more principled solutions, such as aligning data distributions (Xie et al., 2023b) or directly modeling the relationship between mixture and performance. For instance, DoReMi (Xie et al., 2023a) and RegMix (Liu et al., 2025) use proxy models to predict optimal domain weights. However, these methods are designed for and evaluated in different domains within the textual modality, which do not consider the complex cross-modal interactions, and the highly asymmetric computational costs of different modalities, leaving a critical gap in designing principled data mixture strategies for multimodal systems.

**Scaling Laws.** The scaling law of language models which builds the relationship between the performance and model size, dataset size, and compute, have guided LLM development, (Henighan et al., 2020; Kaplan et al., 2020). Recently, the research line of scaling laws has extended to data mixtures. Ye et al. (2025) propose data mixing laws to predict language modeling performance, and Que et al. (2024) design to determine optimal domain ratios during continual pretraining. For MLLMs,

the study of scaling laws is still nascent. While initial works confirm that similar principles apply, they also highlight the additional complexity induced from cross-modal interactions (Aghajanyan et al., 2022; Shukor et al., 2025). However, existing scaling laws for data mixing often assume a scale-invariant optimal ratio, neglecting the joint effects among the multimodal data mixture, model size, and number of training samples to guide efficient MLLM pretraining.

**Omni-Model** aims for general-purpose intelligence, capable of processing and integrating information from text, images, audio, video, and potentially other sensor data within a single, unified architecture. Early examples moving in this direction include models like DeepMind's Gato Reed et al. (2022), which was trained to perform a multitude of tasks from playing games to captioning images and engaging in dialogue. More recent advancements, such as InternOmni (Chen et al., 2024), QwenOmni (Xu et al., 2025), AnyGPT (Zhan et al., 2024), and EMOVA Chen et al. (2025) have pushed the boundaries by demonstrating impressive zero/few-shot capabilities across diverse benchmarks and modalities.

## 3 METHOD

In this section, we first revise the limitations of existing data mixture strategies. We then introduce ModalMix, a novel regression framework designed to optimize multimodal data mixtures by explicitly modeling cross-modal interactions and compute-dependent scaling effects.

### 3.1 PRELIMINARIES

Prior works on data mixture optimization often focus on the uni-modal setting (e.g., balancing different domains within the text modality) (Xie et al., 2023a; Liu et al., 2025). Such methods are inadequate for MLLMs because they address a simplified problem of semantic balancing within a single data format. In contrast, MLLM training is a far more complex challenge of cross-modal alignment, requiring the model to bridge the "modality gap" (Liang et al., 2022) between heterogeneous data structures like pixels, tokens, and waveforms. This alignment process creates intricate inter-dependencies, where the mixture of modalities directly impacts learning dynamics. As empirically shown in Figure 1, for each studied modality, its training loss can vary significantly when the data proportions of the other two modalities vary. In addition, we also observe modality synergy and conflict, where the image-text loss and language loss decline significantly faster when the proportion of speech data increases, indicating the synergy effect between image and speech data, as well as text and speech data. This proves that multimodal learning is not a simple union of independent learning processes, but a complex optimization of interacting systems, a phenomenon that intra-modal methods fail to address.

Another critical limitation of prior work is the implicit assumption that the optimal data mixture $\mathbf{r}^*$ is static and scale-invariant (Liu et al., 2025), overlooking the phenomenon that the total computational budget that comprises model size $N$ and number of samples $D$ significantly impacts learning dynamics. As models scale, their capacity to absorb information from different modalities changes, and the relative value of each data type evolves. As shown in Figure 2a, a data mixture optimized for a 2M-sample budget is suboptimal for a 5M-sample budget, and vice versa. This confirms that the optimal data allocation is inherently compute-dependent, a non-negligible aspect overlooked by previous data mixing strategies.

### 3.2 MODALMIX: COMPUTE-AWARE MIXTURE OPTIMIZATION

To address the challenges of cross-modal interactions and scale-dependence, we introduce ModalMix, a regression-based framework that captures the relationship between modality-specific loss, model scale, compute, and the data mixture itself.

Our framework is applied to a standard omni-modal architecture where inputs from different modalities are mapped into a shared embedding space and processed by a single autoregressive language model. Specifically, considering an MLLM, visual inputs are passed through a vision encoder and then projected into the LLM's embedding space via a trainable vision projector. For speech inputs, the raw audio is tokenized into a sequence of acoustic tokens that undergo a tokenization process through a speech-to-unit module. These newly created tokens are then incorporated into the LLM's

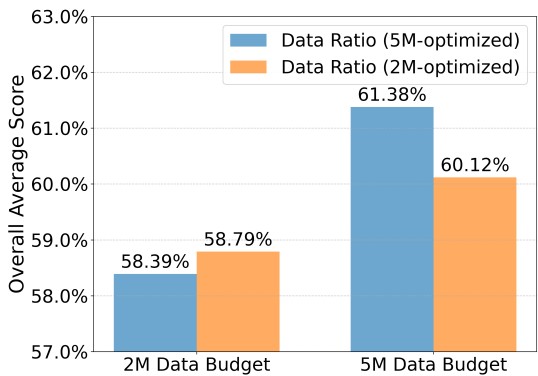
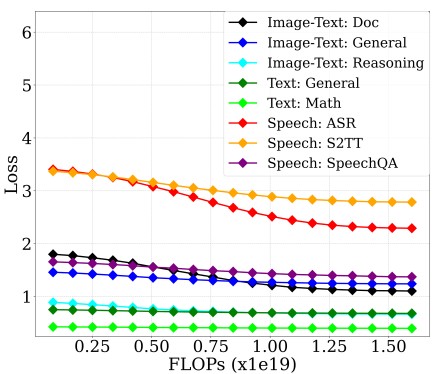

(a) Average score of 17 benchmarks for models trained on data mixture determined by the proposed ModalMix with 5M and 2M total omni-modal data samples, respectively.

(b) Training loss of different modalities and tasks with increasing computational budget measured by FLOPs ($\times 1e19$).

Figure 2: (a) Scale matters in optimal mixture: models achieve higher performance when trained with a data mixture optimized for their specific computational budget (e.g., the 5M-optimized mix on 5M data), confirming that the optimal ratio is not static across scales. (b) Different modalities exhibit distinct learning dynamics: the speech loss saturates quickly after rapid initial gains, while the image-text and text losses show a slower, more sustained decrease. This indicates that the optimal data mixture should be dynamically adjusted over training.

vocabulary, and their embeddings are designed to be dimensionally consistent with the existing text embeddings. As a result, this unified representation allows inputs from all modalities to be combined into a single, interleaved input sequence.

The core design of ModalMix is a scaling law that predicts the loss $L_i$ for each modality $i$ as a function of model size $N$, number of samples $D$, and the multimodal data mixture vector $\mathbf{r} = [r_1, \ldots, r_M], \sum_{i=1}^{M} r_i = 1$ over $M$ modalities. Specifically, we extend the language model scaling law formulation introduced in (Kaplan et al., 2020) to the multimodal context as:

$$L_i(N, D, \mathbf{r}) = E_i + \frac{A_i}{N^{\alpha_i}} + \frac{B_i}{D^{\beta_i}} + C_i \cdot \exp\left(-\sum_{j=1}^{M} \gamma_{ij} r_j\right), \quad \text{s.t. } L_i(N, C, \mathbf{r}) \leq (1+\epsilon) L_i^{\text{lb}}(N, C) \quad (1)$$

where $E_i, A_i, B_i, \alpha_i, \beta_i$ are standard scaling parameters representing the irreducible loss (Kaplan et al., 2020), model size effect, and data effect for modality $i$. The final term is our key innovation for capturing cross-modal dynamics, where $C_i$ is a scaling coefficient, and $\gamma_{ij}$ is a learnable interaction parameter that quantifies the effect of data from modality $j$ on the loss of modality $i$. A positive $\gamma_{ij}$ indicates synergy, meaning that data from modality $j$ helps the learning of modality $i$, while a negative value suggests conflict. This unified formulation allows us to predict the performance of any modality under custom configurations of model scale, compute, and data mixture without running expensive training experiments.

**Estimation of Scaling Parameters.** The parameters of Eq. (1) are determined by fitting the model to empirical data from extensive experimental runs. In these experiments, we systematically varied model size $N$, number of samples $D$, and data mixture $\mathbf{r}$, recording the resulting modality-specific losses ($L_i$). We then estimated the parameters by minimizing the Huber loss (Huber, 1992) between the predicted and observed losses using an L-BFGS optimizer (Liu & Nocedal, 1989), which yields a fully calibrated predictive model capable of forecasting loss for arbitrary training configurations. More details can be found in Section 4.

**Optimization of the Data Mixture.** After we obtain the predictive model in Eq. (1), we can search for the optimal data mixture $\mathbf{r}^*$. However, simply minimizing the total predicted loss without any constraints often leads to a "winner-takes-all" scenario, where the model over-specializes in easy-to-learn modalities, i.e., all the training samples are selected from the simplest modality. To ensure balanced capabilities, we augment this optimization with a crucial constraint, requiring that the loss for each modality $i$ does not significantly exceed its achievable minimum. Formally, for

$i$-th modality, we enforce $L_i(N, C, \mathbf{r}) \leq (1 + \epsilon)L_i^{\text{lb}}(N, C)$, where $L_i^{\text{lb}}$ is the corresponding predicted minimum loss, and $\epsilon$ is a small tolerance margin. In this work, we set $\epsilon = 0.1$. This lower bound can be approximated by applying Eq. (1) with a unimodal data mixture (i.e., $r_i = 1$). This principled constraint prevents any single modality from being sacrificed, allowing ModalMix to find a data mixture that is not only efficient but also fosters well-rounded capabilities.

# 4 EXPERIMENTS

## 4.1 EXPERIMENTAL CONFIGURATION

**Model Architecture.**   We adopt LLaVa (Liu et al., 2023) as the base architecture to handle visual and textual inputs. It integrates a pre-trained vision encoder, an adapter, and a language model. This design aligns visual features with the language model's embedding space via the adapter, retaining the LLM's generation capabilities while extending it to process visual inputs. In practice, we utilize the Qwen-2.5 (Yang et al., 2024) model families as base LLMs, where three configurations (0.5B, 1.5B, 3B) are provided to meet different computational budget requirements. QwenViT (Bai et al., 2025) acts as the visual encode with pixel constraints set to fixed 802,816, and a 2-layer MLP visual projector is used in conjunction. As for the speech part, we adopt SPIRAL (Huang et al., 2022) architecture for the speech encoder to capture phonetic and tonal information, which is then discretized by the finite scalar quantization-based quantizer (Mentzer et al., 2023). The speech codebook has a size of 4,096 with a sample rate of 25 tokens per second. After discretization, the speech modality is integrated into LLMs by concatenating the text vocabulary with the speech codebook.

**Training Datasets.**   We apply a two-stage training with both open-source and in-house datasets. In the first stage, we freeze the language backbone and train the other parameters with image and speech data. Specifically, the visual projector is trained for vision-language alignment on the LCS-558K dataset (Liu et al., 2023). In addition, we train the speech codebook on the Automatic Speech Recognition (ASR) task with 525K data, mainly sampled from AISHELL (Du et al., 2018) and LibriSpeech (Panayotov et al., 2015), enabling the speech codebook to achieve basic understanding of speech data. After that, we train with data from three different modalities: image-text, pure text, and speech data. which is further divided according to tasks into: natural image understanding (General), document understanding (Doc), and image reasoning (Reasoning), text understanding (General), and text reasoning (Reasoning), automatic speech recognition (ASR), speech-to-text translation (S2TT), and speech question answering (SpeechQA). To enable processing by a single autoregressive model, we transform data instances across all modalitie into a unified conversational format as a sequence of *{"role": "user", "content": <query>}* and *{"role": "assistant", "content": <response>}* pairs, which accommodates both single-turn and multi-turn interactions.

**Training Setup.**   Following Chinchilla (Hoffmann et al., 2022), we fix the model size and vary the mixtures of training data. We conduct experiments on 27 different mixture configurations across 3 modalities. Specifically, the mixture configurations of image-text, pure text and speech are {[8,1,1], [7,2,1], [7,1,2], [6,3,1], [6,1,3], [6,2,2], [1,8,1], [2,7,1], [1,7,2], [1,6,3], [3,6,1], [2,6,2], [1,1,8], [2,1,7], [1,2,7], [3,1,6], [1,3,6], [2,2,6], [5,1,4], [1,5,4], [1,4,5], [5,2,3], [2,5,3], [2,3,5], [4,2,4], [2,4,4], [4,4,2]}. For each mixture configuration, we train with 2M total training samples, and save the performance at 20 intermediate checkpoints sampled uniformly during the training process. We fix training hyper-parameters for all experiments. We adopt a global batch size of 128, and a base learning rate of 2e-5 (with the vision encoder fine-tuned at 2e-6). The training employs a cosine learning rate decay scheme. The maximum sequence length is set as 8192.

**Baselines and Evaluation Metrics.**   We compare the proposed ModalMix against three distinct data mixture baselines: (i) Uniform Mixture strategy, where the number of training samples is the same for each modality; (ii) RegMix (Liu et al., 2025), an intra-modal method that relies on training numerous small proxy models to find an optimal static mixture regardless of different model sizes and the number of training samples; and (iii) M2-Omni (Guo et al., 2025) dynamically adjusts the data mixture during training by estimating the learning speed of each modality by periodically calculating its convergence slope. Appendix B.1 provides implementation details for each baseline. We comprehensively evaluate the effectiveness of our proposed ModalMix across three modalities and seventeen tasks in Table 1.

Table 1: Overview of the 17 downstream evaluation benchmarks used in our experiments.

| Modality | Task | Benchmark | Metric |
|---|---|---|---|
| Image-Text | Doc
General
Reasoning | ChatQA, DocVQA, InfoVQA
MME, TextVQA, Textcaps
MathVista, Mathverse, AI2D | Accuracy
Score/Accuracy, ROUGH-L
Accuracy |
| Text | General
Reasoning | InEval
GSM8K, GPQA | Accuracy
Accuracy |
| Speech | ASR
SpeechQA | AISHELL2, LibriSpeech-Test-Clean, LibriSpeech-Test-Other
WebQ, LLamaQ | CER & WER
Accuracy |

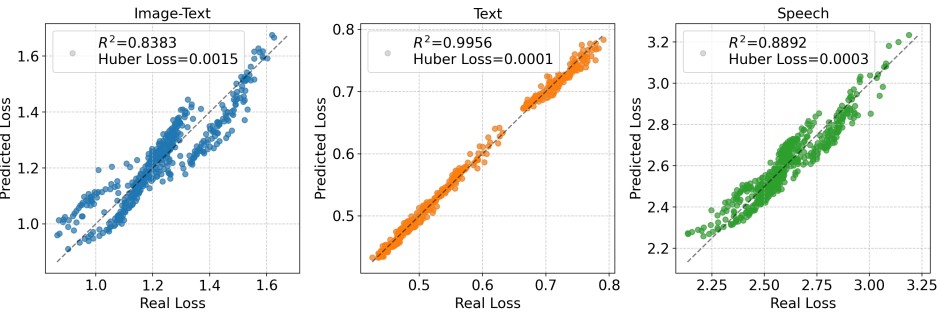

Figure 3: The predicted loss versus empirical loss of different modalities using the proposed ModalMix. Each point represents a model checkpoint under a specific training step and data mixture. The dashed line indicates the $y = x$ identity line, representing a perfect prediction.

## 4.2 ESTIMATION OF THE SCALING PARAMETERS OF MODALMIX

To empirically validate the efficacy of our proposed ModalMix, we first fit the scaling parameters in Eq. (1) across varying model sizes, number of samples, and data mixtures. Specifically, we generate data points for fitting from extensive experiments on (i) three sizes of language backbones (i.e., 0.5B, 1.5B, and 3B); (ii) 27 different data mixture configurations; and (iii) training loss values at 20 intermediate checkpoints of each training run. As a result, we curate a dataset of $3 \times 27 \times 20 = 1,620$ experimental runs, with each data point representing a unique tuple of (model size $N$, training samples $D$, data mixture $\mathbf{r}$) and corresponding loss of image-text, text, and speech ($L_{it}, L_t, L_s$, respectively).

Based on these samples, we then optimize the scaling parameters of the ModalMix by minimizing the Huber loss (Huber, 1992) between the predicted and empirical losses. We use Pearson coefficient (Benesty et al., 2009) $R^2$ to measure the quality of the fitting result. From Figure 3, the data points are clustered around the diagonal $y = x$, indicating a strong linear correlation between the predicted and real losses for all three modalities. This observation substantiates the capability of our ModalMix to accurately model the training dynamics of different modalities.

From a quantitative point of view, the performance of the fitting is similarly compelling for all three modalities. For the text modality, the predictive model achieved a coefficient of determination ($R^2$) of 0.9960, indicating a near-perfect fit. Commendable $R^2$ values were also obtained for the speech and image-text modalities, at 0.8878 and 0.8385, respectively. Moreover, the consistently low Huber loss across all modalities further attests to the predictive precision of the model. Collectively, these findings provide robust empirical evidence that the ModalMix function acts as an effective tool for performance prediction, offering reliable loss forecasts for omni-modal pre-training under various model sizes and data mixture, establishing a solid foundation for the subsequent application of these laws to data mixture optimization and compute allocation strategies.

**Visualization of Cross-modal Interactions.** To quantify inter-modal dynamics, we analyzed the Pearson correlation of the final modality-specific losses from our 27 experimental runs (Figure 4a). The results show a consistently positive correlation across all modality pairs, confirming a synergistic relationship where co-training provides mutual benefits. The strongest effect is observed from speech to both image-text ($\rho = 0.63$) and text ($\rho = 0.50$), which aligns with our finding that more speech data improves the convergence of other modalities (Figure 1). Furthermore, the relationships are asymmetric. For instance, the influence of speech on other modalities is stronger than their influence back, in contrast to the more balanced relationship between text and image-text ($\rho \approx 0.45$). These findings demonstrate the complex but predominantly positive nature of cross-modal interactions.

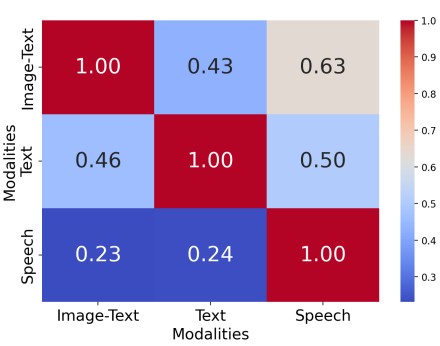
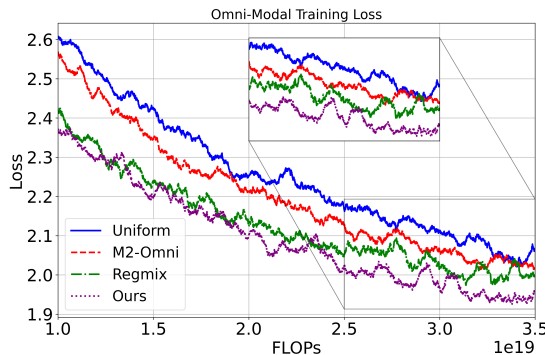

(a) Correlation matrix of final training losses between modalities. Warmer colors (red) indicate a stronger positive correlation, while cooler colors (blue) indicate a weaker correlation.

(b) Omni-modal training loss of different data mixture methods on a 1.5B language backbone. Our method converges to the lowest loss $\frac{3.5}{2.5} = 1.4\times$ faster than the second best method Regmix.

Figure 4: Correlation matrix of final training losses between modalities with ModalMix, and loss comparison across different data mixture methods.

Table 2: Performance comparison of different data mixture methods with omni-training on a 1.5B language backbone. Our propsoed ModalMix not only has the best overall performance on multi-tasks in all three modalities, but also achieves balanced capabilities across modalities. The highest and second-highest scores of each method are respectively highlighted in red and blue.

| Modality | Task | Benchmark | Metric | Method | | | |
|---|---|---|---|---|---|---|---|
| | | | | Uniform | RegMix | M2-Omni | Ours |
| Image-Text | Doc | Chartqa | Acc. | 72.4 | 74.2 | 72.4 | 72.5 |
| | | DocVQA | Acc. | 89.0 | 89.7 | 89.3 | 89.9 |
| | | InfoVQA | Acc. | 63.1 | 63.5 | 61.3 | 61.7 |
| | General | MME | Score | 1955.8 | 1868.7 | 1885.6 | 1969.5 |
| | | TextVQA | Acc. | 70.2 | 71.4 | 70.0 | 70.8 |
| | | Textcaps | ROUGE-L | 51.3 | 52.1 | 51.7 | 52.2 |
| | Reasoning | MathVista | Acc. | 50.7 | 52.2 | 51.3 | 51.7 |
| | | Mathverse | Acc. | 19.7 | 17.3 | 18.8 | 19.2 |
| | | AI2D | Acc. | 69.0 | 69.5 | 68.7 | 69.6 |
| | **Rank** | | **Avg.** | 3.06 | 1.94 | 3.28 | 1.72 |
| Text | General | IFEval | Acc. | 37.7 | 37.1 | 38.0 | 39.0 |
| | Reasoning | GSM8K | Acc. | 66.0 | 57.3 | 65.3 | 65.4 |
| | | GPQA | Acc. | 25.6 | 23.5 | 25.3 | 24.6 |
| | **Rank** | | **Avg.** | 1.67 | 4.00 | 2.33 | 2.00 |
| Speech | ASR | AISHELL2 | CER | 8.3 | 6.1 | 6.5 | 6.6 |
| | | LibriSpeech-Test-Clean | WER | 5.2 | 4.2 | 4.5 | 4.4 |
| | | LibriSpeech-Test-Other | WER | 8.6 | 7.6 | 7.7 | 7.9 |
| | SpeechQA | WebQ | Acc. | 40.6 | 43.1 | 43.5 | 43.7 |
| | | LLamaQ | Acc. | 67.5 | 71.0 | 72.0 | 71.3 |
| | **Rank** | | **Avg.** | 4.00 | 1.80 | 2.00 | 2.20 |
| **Average Rank** | | | | 2.91 | 2.58 | 2.54 | 1.97 |

## 4.3 COMPARISON WITH OTHER DATA MIXTURE BASELINES

In this section, we compare our proposed ModalMix with other data mixture methods on a 1.5B language backbone. Results in Figure 4b and Table 2 validate the efficacy of our approach with a 1.5B language backbone: data mixtures determined by ModalMix yield the lowest training loss and fastest convergence among all the compared methods. Across 17 downstream tasks (Table 2), our method achieves the best overall average rank (1.97) across all modalities, outperforming other baselines. Notably, it demonstrates balanced capability: ranking 1st in image-text (1.72) while maintaining top-tier ranks in text (2.00) and speech (2.20), indicating that ModalMix mitigates inter-modal conflicts for synergistic learning.

Table 3: Generalization results on unseen 7B models. The data mixture of ours is predicted based on the data fitting on 0.5B, 1.5B and 3B models. The highest and second-highest scores of each method are respectively highlighted in red and blue.

| Modality | Task | Benchmark | Metric | Method | | | |
| --- | --- | --- | --- | --- | --- | --- | --- |
| | | | | Uniform | RegMix | M2-Omni | Ours |
| Image-Text | Doc | Chartqa | Acc. | 80.1 | 79.6 | 80.4 | 80.4 |
| | | DocVQA | Acc. | 94.6 | 95.5 | 95.3 | 95.0 |
| | | InfoVQA | Acc. | 76.2 | 77.3 | 76.3 | 76.6 |
| | General | MME | Score | 2183.7 | 2195.3 | 2221.1 | 2203.6 |
| | | TextVQA | Acc. | 53.3 | 52.6 | 53.0 | 53.0 |
| | | Textcaps | ROUGE-L | 75.7 | 75.4 | 76.2 | 76.8 |
| | Reasoning | MathVista | Acc. | 61.0 | 61.4 | 61.9 | 62.3 |
| | | Mathverse | Acc. | 27.0 | 26.4 | 29.3 | 27.0 |
| | | AI2D | Acc. | 78.8 | 78.3 | 78.1 | 78.6 |
| | **Rank** | | **Avg.** | 2.94 | 3.00 | 2.11 | 1.94 |
| Text | General | IFEVAL | Acc. | 41.5 | 40.1 | 40.4 | 41.9 |
| | Reasoning | GSM8K | Acc. | 62.9 | 63.2 | 62.9 | 63.5 |
| | | GPQA | Acc. | 25.1 | 25.4 | 24.7 | 25.5 |
| | **Rank** | | **Avg.** | 2.83 | 2.67 | 3.50 | 1.00 |
| Speech | ASR | AISHELL2 | CER | 6.1 | 4.9 | 5.8 | 5.6 |
| | | LibriSpeech-Test-Clean | WER | 4.0 | 3.4 | 3.7 | 3.7 |
| | | LibriSpeech-Test-Other | WER | 7.3 | 6.7 | 7.2 | 7.0 |
| | SpeechQA | WebQ | Acc. | 76.3 | 74.7 | 77.0 | 79.0 |
| | | LLamaQ | Acc. | 50.7 | 44.7 | 48.5 | 49.4 |
| | **Rank** | | **Avg.** | 3.20 | 2.20 | 2.70 | 1.90 |
| **Average Rank** | | | | 2.99 | 2.62 | 2.77 | 1.61 |

## 4.4 GENERALIZATION TO LARGER MODELS

A critical test for any scaling law is its ability to extrapolate beyond the data used for fitting. To further test the generalization ability of the proposed ModalMix to unseen scales, we apply it to a 7B model. Leveraging loss functions fitted on 0.5B/1.5B/3B models, we predict optimal modality mixture $\mathbf{r}^*$ for 7B training. As shown in Table 3, our method achieves the best overall average rank (1.61) on 7B, outperforming baselines across image-text, text, and speech tasks. This confirms ModalMix's strong generalization to larger, unseen model scales, where fitted scaling laws translate to superior downstream performance.

## 4.5 DISCUSSIONS

Having established the predictive power and parameterized form of the ModalMix, we now explore its practical applications.

**The Modality-Specific Loss Contour with Data Mixture.** A key application of ModalMix is to visualize the complex data mixture landscape, enabling the identification of optimal configurations without exhaustive grid searches. Figure 5 illustrates this by plotting the predicted total loss. The figure contrasts two strategies: a naive, unconstrained optimization (red star) that simply minimizes the total loss, versus our proposed constrained approach. While the unconstrained method risks creating an imbalanced model, our approach incorporates a loss lower-bound constraint to find a solution that effectively balances overall performance without sacrificing any single modality. This demonstrates how ModalMix navigates the trade-offs in multimodal data allocation. (See Appendix C.3 for detailed landscapes.)

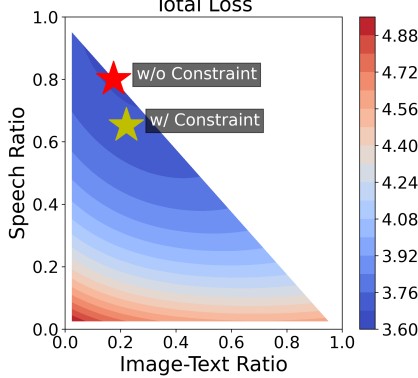

Figure 5: Visualization of the global loss contours across data mixtures of image-text, text, and speech modalities.

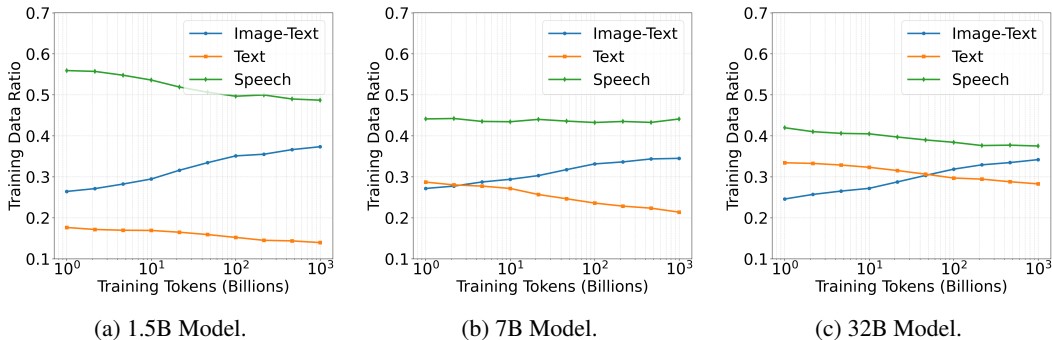

(a) 1.5B Model.  (b) 7B Model.  (c) 32B Model.

Figure 6: Optimal data mixture predicted by ModalMix, with varying model sizes and number of data samples.

**Balance between Textual Intelligence and Capabilities of Other Modalities.**  A key challenge in omni-modal development is to enhance new modal capabilities without degrading the base LLM's core textual intelligence. The constrained optimization formulation of ModalMix in Section 3.2 is designed to address this trade-off. The results of the 1.5B and 7B langauge bakcbones are shown in Table 2 and Table 3, respectively. At 1.5B, baselines like RegMix excel in speech (rank 1.80) but fail in text (rank 4.00), whereas ModalMix maintains strong text (2.00) while ranking highly in image-text (1.72) and speech (2.20). This balanced performance persists at 7B, where ModalMix achieves the best overall rank, demonstrating its effectiveness in producing well-rounded models without sacrificing linguistic strength.

**Optimal Data Mixtures Evolve with Computational Budget.**  Our ModalMix framework reveals that the optimal data mixture is not static but should be adjusted for different model scales. For instance, the 1.5B model requires a "bootstrapping" curriculum (starting with 55% speech), while the larger 7B model adopts a more balanced approach from the outset (Figures 6a-6b). Extrapolating to 32B parameters with ModalMix (Figure 6c) predicts a rising demand for text (>30%) as speech declines, suggesting that core reasoning eventually supersedes explicit cross-modal alignment. This trajectory points to a fundamental phase transition in learning strategy: from signal-level bootstrapping (small scale), to balanced alignment (medium scale), and finally to language-centric abstract reasoning (extreme scale). Ultimately, scaling does not just improve performance—it changes the nature of learning, making a compute-aware framework like ModalMix essential for devising optimal data strategies. More details can be found in Appendix C.2.

**Correlation Between Training Loss and Test Accuracy.**  To understand if pretraining loss is a reliable proxy for generalization, we analyze the correlation between the final training loss of each modality and its downstream task performance. Our analysis reveals that pretraining loss is an unreliable proxy for downstream performance in a modality-dependent manner. While strongly correlated for perceptual tasks (image-text and speech), the relationship vanishes for complex text reasoning, underscoring the need for a more nuanced optimization approach. More details can be found in Appendix C.4.

## 5 CONCLUSIONS

In this work, we introduce ModalMix, which is a scaling framework for omni-modal models that first optimizes data mixtures by modeling both cross-modal interactions and compute-dependent dynamics. Our approach enables accurate loss prediction across model scales, number of samples, and data mixture. Empirical results show that ModalMix accelerates the pre-training convergence by 1.4× and boosts average rank across 17 downstream tasks in three modalities by 47%, while balancing performance across modalities. The proposed ModalMix reveals the evolution trend of compute-optimal data mixture, and offers actionable guidance for resource-aware data scheduling.

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

## A STATEMENT ON LARGE LANGUAGE MODELS (LLMS) USAGE

In the preparation of this paper, we utilized Large Language Models (LLMs) solely for the purpose of grammatical polishing and text refinement of the manuscript content. Specifically, the LLMs were only used to optimize the clarity, fluency, and grammatical accuracy of the written text.

All content polished by LLMs underwent thorough manual review and verification by the authors. We carefully checked the polished text to ensure its consistency with the original research intent, accuracy of scientific facts, and compliance with academic integrity standards. We confirm that we take full responsibility for all contents of the paper under our names, including the parts that underwent LLM-assisted grammatical polishing.

## B  EXPERIMENTS

### B.1  IMPLEMENTATION DETAILS OF BASELINES

This section provides detailed implementation information for the three baseline methods used in our comparative experiments.

**Uniform Mixture.**  baseline is designed to provide equal exposure to each modality during training, serving as a simple yet crucial reference point. This process ensures that, over the course of training, the model is trained on an equal number of samples from each of the three modalities, regardless of the original size of their respective datasets.

**RegMix.**  baseline is a predictive method designed to find a near-optimal, static data mixture before conducting the main, large-scale training. To achieve this, we trained several 1.5B parameter models, each using a different, predefined data mixture, denoted as $x$. After training, we evaluated each model on a held-out validation set and recorded its mean loss, which served as the optimization target, $y$. Using this collection of (mixture, validation loss) pairs, we trained a RandomForestRegressor (Biau, 2012) to predict the loss $y$ for any given input $x$, instead of LightGBM (Ke et al., 2017) as used in the original RegMix paper, due to the heavy over-fitting. Once the regressor was trained, we initiated an optimization phase by generating 100,000 random candidate data distributions and used the regressor to predict the validation loss for all candidates and selected the top-k (where k=128) distributions that yielded the lowest predicted losses. The final, optimal data mixture was determined by averaging these top 128 distributions, and this static mixture was then used for the entire training run of the RegMix baseline model.

**M2-Omni.**  is a dynamic, online baseline that continuously adapts the data mixture during training by monitoring the model's learning progress in each modality. We also introduce a baseline inspired by the training dynamics of M2-omni. The creation of this baseline is a two-stage process. First, we train a 1.5B parameter model using a uniform data mixture, where each modality is sampled equally. During this initial training run, we monitor the learning progress of each modality by periodically calculating its training loss. Following the methodology for the dynamic adaptive balance strategy in M2-omni, we then use linear regression on the historical losses to compute the convergence slope for each modality. In the second stage, we use these slopes to create a new, fixed sampling distribution for the final baseline model. The sampling weight for each modality is set to be proportional to its observed convergence slope from the initial run, effectively prioritizing modalities that demonstrated faster learning. The final 1.5B baseline model is then trained from scratch using this new, slope-guided data mixture.

## C  SUPPLEMENTARY DISCUSSIONS

### C.1  OPTIMAL COMPUTATION ALLOCATION

A key application of the ModalMix is to guide the strategic allocation of computational resources, moving beyond the limitations of fixed, scale-invariant data mixture strategies. A critical question arises: *Does the optimal data mixture remain constant across different computational budgets?* Our framework addresses this by treating the data mixture selection as a compute-aware optimization problem. Leveraging Eq. (1), which incorporates model size $N$ and number of samples $D$ as variables, we can solve for the data mixture vector $\mathbf{r}$ that minimizes a predefined objective for any given computational budget.

To empirically validate that the optimal allocation is indeed scale-dependent, we conducted an experiment, with results presented in Figure 2a of the main paper. We first used our ModalMix to determine two distinct optimal data mixtures, one tailored for a small data budget (2M samples) and another for a larger budget (5M samples). We then trained models under both budget constraints, using both the matched and mismatched data ratios. The results provide clear evidence for the scale-dependency of optimal data mixtures. When training with a 5M data budget, the model using the 5M-optimized ratio achieved a superior overall score (61.38%) compared to the one using the 2M-optimized ratio (60.12%). Conversely, under the 2M data budget, the 2M-optimized ratio led to

better performance (58.79%) than the 5M-optimized ratio (58.39%). This directly demonstrates that a data mixture optimized for one computational budget can be suboptimal at another. Therefore, for efficient omni-modal training, it is crucial to adapt the data strategy to the available compute. Our ModalMix provides a principled and automated mechanism to achieve this, and the effectiveness of the resulting allocation strategy is further corroborated by the strong downstream performance detailed in Section 4.3.

## C.2 HOW OPTIMAL DATA MIXTURES EVOLVE WITH COMPUTATIONAL BUDGET

Our ModalMix framework reveals that the optimal data mixture is not a static recipe but a dynamic strategy fundamentally reshaped by model scale. By comparing the empirically-derived schedules for 1.5B and 7B models (Figure 6a and Figure 6b), we identify a clear evolution in learning paradigms. The smaller 1.5B model requires a bootstrapping curriculum, initiating with a heavy dependency on speech data ( 55%) before gradually shifting towards complex image-text alignment. In contrast, the larger 7B model adopts a more balanced and accelerated learning path from the outset, processing diverse modalities in parallel thanks to its greater capacity.

Extrapolating these findings using ModalMix's predictive power (Figure 6c) forecasts how these trends will culminate at a typical scale of 32B parameters. The projection indicates a relatively high demand for text data, while the need for speech continues to decline. Intriguingly, the ratio of image–text data is predicted to increase, suggesting that its utility is rising in response to the growing demand for image–text alignment.

This projected evolution signifies a fundamental phase transition in how models learn as they scale. The strategy shifts from signal-level bootstrapping at a small scale, to balanced cross-modal alignment at a medium scale, and ultimately towards text-centric abstract reasoning at an extreme scale. Here, a deep language backbone becomes the central engine for interpreting all other modalities. Therefore, scaling compute does not just enhance performance; it changes the nature of learning itself, necessitating a compute-aware framework to devise truly optimal data strategies.

## C.3 MORE DETAILS ABOUT PREDICTED LOSS LANDSCAPES

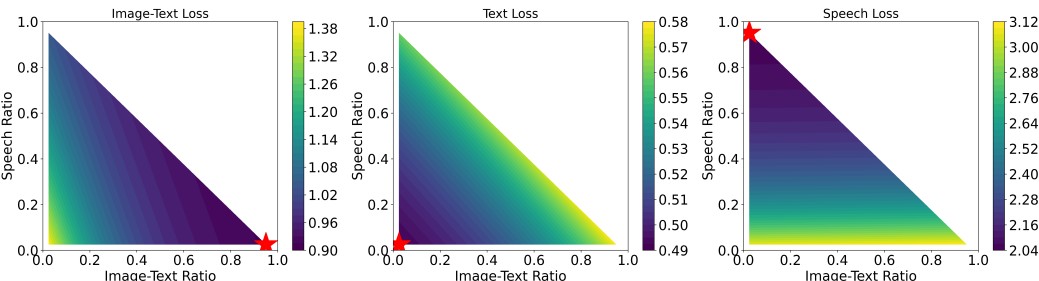

Figure 7: Predicted loss contours for each modality as a function of the data mixture ratio. The x-axis represents the ratio of image-text data, and the y-axis represents the ratio of speech data. The ratio of text data is implicitly determined as $(1 - r_{it} - r_s)$. Darker regions indicate a lower predicted loss. The red star in each plot marks the data mixture that yields the minimum loss for that specific modality.

Figure 7 visualizes these loss landscapes for a fixed model size and total number of samples. Each subplot corresponds to a single modality, illustrating its predicted final loss across all possible three-way data splits. The contours reveal distinct optimal data mixtures for each modality. For instance, to minimize the image-text Loss (left subplot), the model benefits most from a data mixture heavily dominated by image-text data itself, as indicated by the red star located at the bottom-right corner of the simplex. Conversely, minimizing the text Loss (center subplot) requires a mixture almost entirely composed of pure text data (bottom-left corner). Similarly, the optimal point for speech Loss (right subplot) is found at a high proportion of speech data (top-left corner).

These visualizations not only pinpoint the optimal data recipe for specializing in a single modality but also allow for a nuanced understanding of the trade-offs involved. For example, one can observe how increasing the speech data ratio (moving upwards along the $y$-axis) gradually increases the predicted

loss for both image-text and text tasks. This predictive capability is crucial for making informed decisions about data mixture to achieve a desired balance of capabilities in the final model.

### C.4   TRAINING LOSS AS A PROXY FOR DOWNSTREAM CAPABILITY IS HIGHLY MODALITY-DEPENDENT

To understand the complex relationship between pretraining objectives and generalization, we analyze how the final training loss for each modality correlates with performance on its corresponding downstream tasks. We compare models trained with varied data mixtures but identical model sizes and total compute. The results, visualized in Figures 8 to 9, reveal that the utility of training loss as a proxy for downstream capability is highly modality-dependent.

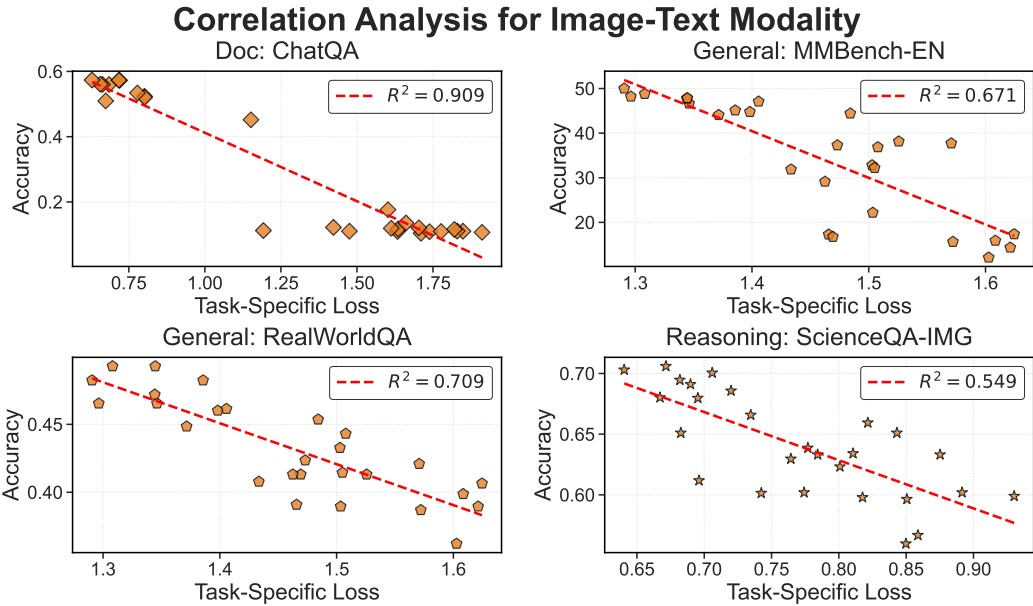

Figure 8: **Image-Text: Strong Correlation.** The relationship between the final image-text training loss and performance on corresponding downstream tasks. For visual-linguistic tasks like document understanding (ChatQA) and general QA (MMBench, RealWorldQA), a lower training loss strongly and consistently correlates with better downstream accuracy. Each point represents a model from a different data mixture experiment.

As shown in Figure 8, we observe a strong and consistent negative correlation between the image-text training loss and performance on downstream tasks. Across benchmarks like document understanding (ChatQA, $R^2 = 0.91$) and general visual reasoning (MMBench, $R^2 = 0.67$; RealWorldQA, $R^2 = 0.71$), a lower final training loss reliably predicts higher accuracy. A similar trend holds for the speech modality (Figure 9). Speech-centric tasks, particularly Automatic Speech Recognition (ASR) on datasets like AISHELL2 ($R^2 = 0.93$) and LibriSpeech ($R^2 > 0.82$), and Speech-to-Text Translation (S2TT) on COVOST2 ($R^2 > 0.79$), exhibit a very strong correlation. This indicates that for perceptual modalities like image-text and speech, the pretraining loss serves as a valid proxy for downstream capabilities.

In stark contrast, the text modality presents a completely different picture, as shown in Figure 10. For complex reasoning and knowledge-intensive tasks such as MMLU ($R^2 = 0.02$), instruction following (IFEVAL, $R^2 = 0.02$), and mathematical reasoning (GSM8K, $R^2 = 0.02$), the correlation between the next-token prediction loss and final accuracy is virtually non-existent. The scattered data points confirm that simply minimizing the text training loss does not guarantee improvements in these sophisticated linguistic abilities.

These findings reveal a critical insight: the utility of training loss as a proxy for downstream performance is highly inconsistent across modalities. While minimizing loss is effective for perceptual tasks (image-text, speech), it is a misleading and unreliable indicator for complex text reasoning.

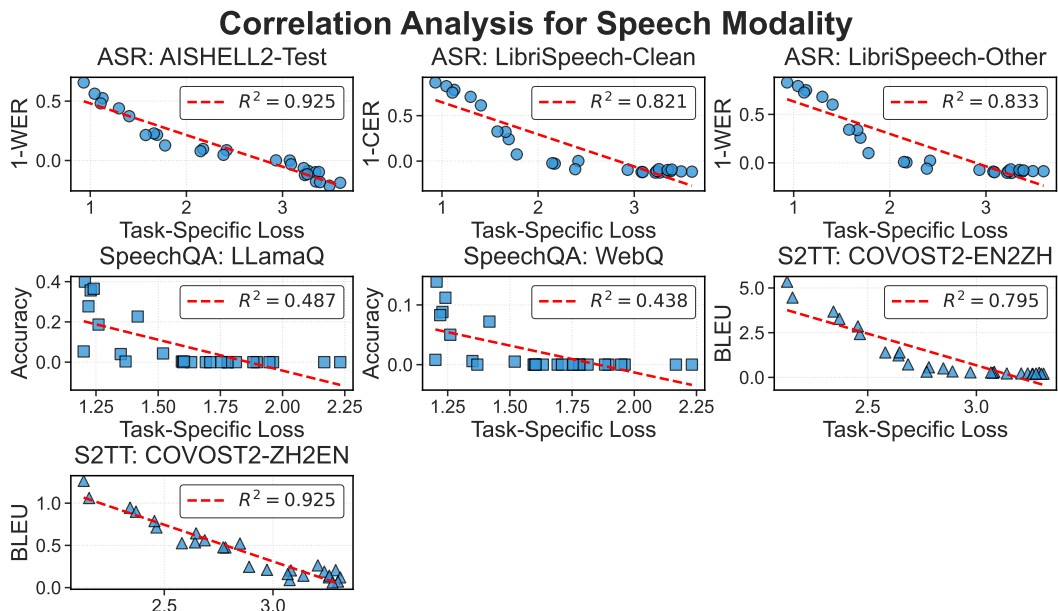

Figure 9: **Speech: Strong Correlation.** The relationship between the final speech training loss and performance on speech-related tasks. Similar to the image-text modality, lower loss is a reliable indicator of improved performance, especially for ASR (e.g., LibriSpeech) and S2TT (e.g., COVOST2), which show very high R² values.

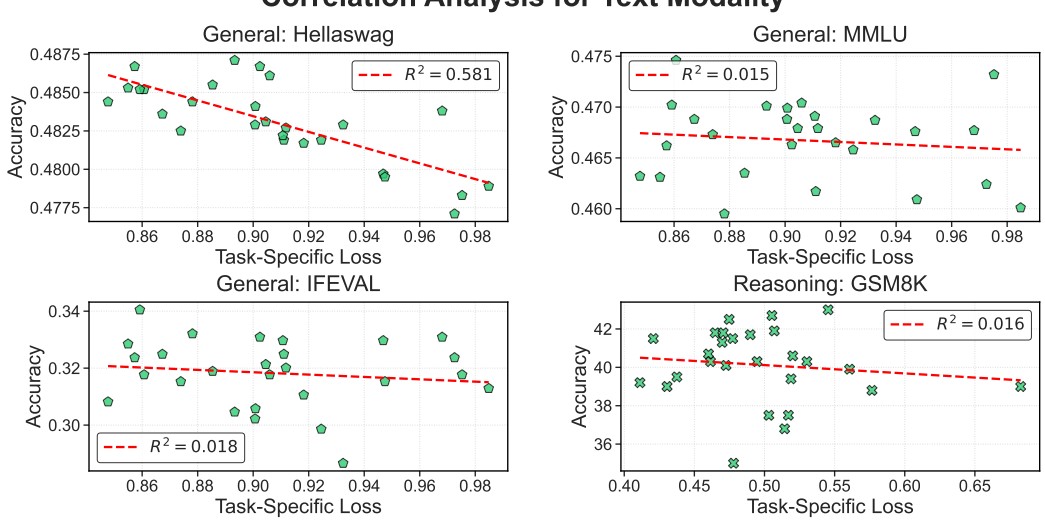

Figure 10: **Text: Weak to No Correlation.** The relationship between the final text training loss and performance on complex text tasks. In stark contrast to other modalities, there is virtually no correlation between the next-token prediction loss and performance on reasoning-heavy benchmarks like MMLU, IFEVAL, and GSM8K.

This divergence demonstrates that a naive optimization strategy that solely aims to minimize a weighted sum of training losses is fundamentally flawed. It underscores the necessity for a more sophisticated framework like our proposed ModalMix, which can intelligently balance performance across modalities instead of being misguided by unreliable loss signals.

