# OpenReview forum: "ModalMix: Optimizing Multimodal Data Mixtures with Compute-Dependent Regression"
_ICLR.cc/2026/Conference — Submitted to ICLR 2026_

### Official Review · Reviewer_ukzq · 2025-10-25

**Soundness:** 3
**Presentation:** 3
**Contribution:** 2
**Rating:** 4
**Confidence:** 3

**Summary:**

This paper introduces ModalMix, a regression-based framework that optimizes multimodal data mixtures by jointly modeling cross-modal interactions and compute-dependent scaling laws. The key idea is to predict the optimal data mixture for any given computational budget, addressing the limitations of static or heuristic data allocation strategies in multimodal learning.

ModalMix learns scaling laws that link model size, number of samples, and modality ratios to modality-specific losses, and it constrains optimization to avoid overfitting to any single modality. Experiments across text, image, and speech modalities (17 downstream tasks) demonstrate 1.4× faster convergence and a 47% improvement in average rank over baselines. The method generalizes well to unseen larger models (e.g., 7B parameters) and offers insights into how optimal mixtures evolve with scale—transitioning from speech-heavy at small scales to text-dominant at large scales.

**Strengths:**

1. The paper clearly identifies and formalizes a critical, under-explored challenge in multimodal learning: the joint optimization of data mixture while accounting for cross-modal dynamics and computational scale. This moves beyond unimodal and static mixture strategies.
2. The scale of the experimental setup (1,620 runs across 3 model sizes and 27 mixtures) is impressive and provides a solid foundation for the model. The results are comprehensive, showing: (1) strong performance gains in convergence speed and downstream task performance；(2) Effective generalization to an unseen, larger (7B) model.

**Weaknesses:**

1. Although the paper emphasizes compute-awareness, it lacks a clear quantitative comparison of the additional cost introduced by ModalMix (e.g., time to fit scaling parameters, overhead from mixture adjustment).
2. The study is confined to text, image-text, and speech. The framework's applicability to other important modalities like video, 3D data, or tabular data remains an open question. A discussion on this limitation and potential for generalization would strengthen the paper.
3. The scaling laws and optimal mixtures are derived and validated within a specific architecture family (LLaVA-like with Qwen backbones). It is unclear how sensitive the findings are to the underlying model architecture (e.g., the design of the vision projector, the speech tokenizer, or the choice of a purely autoregressive objective). The "optimal" mixture might be architecture-specific.
4. The ablation studies focus on predictive performance but do not test robustness to noisy or imbalanced datasets. Real-world multimodal corpora often have high variance in quality and domain coverage.

**Questions:**

1. The paper mentions using L-BFGS for parameter fitting and a search over the mixture space. Could you elaborate on the computational cost of this "outer-loop" optimization process itself, and how it compares to the cost of the training runs used for fitting?
2. How sensitive is the ModalMix predictor to the initial dataset composition used for fitting scaling parameters? Would training on different data sources change the predicted optimal mixtures?

---

> ### Author Response · Authors · 2025-11-21
> **Reply to reviewer ukzq [Part1]**
>
> **Question:** The paper lacks quantitative comparison of ModalMix’s additional costs.
>
> **Response:**
> ModalMix’s computational overhead is minimal compared to multimodal pre-training (which takes days/weeks):
> 1. **Mixture Prediction**: Predicting optimal mixtures for new ($N, D$) via the pre-fitted function is negligible (instantaneous).
> 2. **Training Overhead**: No online cost—mixtures are computed once pre-training and fixed, with no per-step monitoring or dynamic reweighting (unlike M2-Omni, which requires run-specific re-estimation and is noise-sensitive).
>
> These confirm ModalMix’s compute-efficiency, adding minimal overhead to standard pretraining.
>
>
> **Question:** A discussion on limitation and generalization would strengthen the paper.
>
> **Response:**
> We agree and will add a dedicated limitation section. Conceptually, ModalMix is **modality-agnostic**: extending to video, 3D, or tabular data only requires expanding the mixture vector $r$ and fitting new $\gamma_{ij}$ (no framework changes), as the core relies on modality-specific loss $L_i(N, D, r)$ and cross-modal interactions. Notably, our current image-text branch indirectly includes video-like (multi-frame images) and tabular/structured (document understanding) data. We will clarify this generalization potential and frame the current scope (text, image-text, speech) as a starting point for broader multimodal extension.
>
>
> **Question:** The scaling laws and optimal mixtures are derived for a specific LLaVA-like (Qwen backbone) architecture.
>
> **Response:**
> We do not claim universal applicability—our work focuses on a **widely used LLaVA-like architecture**, a standard design for modern MLLMs. Inspired by OpenAI’s scaling law findings [5], functional forms are stable once total compute is fixed (width/depth variations have minor impacts, with model size and data volume as key drivers). Thus, our conclusions characterize compute-optimal mixtures for this standard LLaVA-like stack.
>
>
> **Question:** Ablation studies focus on predictive performance but lack robustness tests for noisy/imbalanced datasets—common in real-world multimodal corpora.
>
> **Response:**
> Our experiments use **real-world multimodal data** (inspired by EMOVA [3]) with natural noise and imbalance, aggregating diverse sources with varying quality/domain coverage. All scaling-law fitting and mixture optimization are conducted under realistic distributional conditions.
>
> We agree targeted stress tests (e.g., systematic label noise, extreme skew) would strengthen validation. We will clarify that current results reflect real-world data characteristics and frame robustness-focused experiments as a key future direction.
>
>
> **Question:** Could you elaborate on the computational cost of the "outer-loop" optimization (L-BFGS for parameter fitting, mixture space search) and its comparison to training run costs?
>
> **Response:**
> We fit Eq. (1) in log-loss space via L-BFGS (max 1,000 iterations), minimizing Huber loss (δ=1.0) and selecting parameters with the highest $R^2$. Running on a 12-core Intel CPU, the full process (all modalities) takes <1 minute—trivial compared to the hours/days of GPU time per pretraining run (used to generate checkpoints). We will add these details to the main text/appendix for transparency and reproducibility.

---

> ### Author Response · Authors · 2025-11-21
> **Reply to reviewer ukzq [Part2]**
>
> **Question:** How sensitive is ModalMix to the initial dataset composition for fitting scaling parameters?
>
> **Response:**
> ModalMix is trained on a **real-world, multi-domain multimodal data pool** (aligned with LLM scaling law research [1][2][3][4][5], which prioritizes data scale over explicit distribution assumptions). We sample data per experiment based on compute budget without imposing artificial distribution constraints—this design mirrors practical MLLM pretraining environments.
>
> | Modality   | Domains                                                                 | Datasets                                                                 |
> | :--------- | :---------------------------------------------------------------------- | :----------------------------------------------------------------------- |
> | Image-Text | General, OCR, Document, Chart, Screen, Math, Science                     | ShareGPT-4o, LVIS-Instruct4V, LLaVAR, MAVIS, G-LLaVA, ...                |
> | Text       | General, Reasoning, Math                                                | Magpie, ShareGPT4, MathPlus, ...                                         |
> | Speech     | ASR, SpeechQA, S2TT                                                     | AISHELL2, LibriSpeech, Emova, ...                                        |
>
> The diverse, naturally distributed corpus ensures robustness to initial dataset composition. We will add detailed descriptions of the pretraining data pool in the revised manuscript for transparency.
>
> [1] Hoffmann, J., Borgeaud, S., Mensch, A., Buchatskaya, E., Cai, T., Rutherford, E., ... & Sifre, L. (2022). Training compute-optimal large language models. arXiv preprint arXiv:2203.15556.
>
> [2] Shukor, M., Bethune, L., Busbridge, D., Grangier, D., Fini, E., El-Nouby, A., & Ablin, P. (2025). Scaling laws for optimal data mixtures. arXiv preprint arXiv:2507.09404.
>
> [3] Chen, K., Gou, Y., Huang, R., Liu, Z., Tan, D., Xu, J., ... & Xu, H. (2025). Emova: Empowering language models to see, hear and speak with vivid emotions. In Proceedings of the Computer Vision and Pattern Recognition Conference (pp. 5455-5466).
>
> [4] Ye, J., Liu, P., Sun, T., Zhan, J., Zhou, Y., & Qiu, X. (2024). Data mixing laws: Optimizing data mixtures by predicting language modeling performance. arXiv preprint arXiv:2403.16952.
>
> [5] Kaplan, J., McCandlish, S., Henighan, T., Brown, T. B., Chess, B., Child, R., ... & Amodei, D. (2020). Scaling laws for neural language models. arXiv preprint arXiv:2001.08361.

---

### Official Review · Reviewer_3KNk · 2025-10-26

**Soundness:** 2
**Presentation:** 3
**Contribution:** 3
**Rating:** 6
**Confidence:** 3

**Summary:**

This paper addresses the critical and practical problem of optimizing data mixtures for large-scale multimodal pre-training. The authors propose ModalMix, a novel framework that extends scaling laws to the multimodal domain. The core of ModalMix is a regression model that predicts modality-specific training losses as a function of model size, data volume, and the data mixture ratio. By fitting this model on extensive experimental data, the authors obtain a predictive regressor that can efficiently find a compute-aware optimal data mixture. Empirical results on a comprehensive suite of 17 downstream tasks show that models trained with the ModalMix-derived mixture achieve significantly faster convergence and superior overall performance compared to strong baselines.

**Strengths:**

**Practical Significance and Strong Empirical Results**: The paper tackles a highly relevant, real-world problem in large-scale MLLM. The proposed framework demonstrates substantial gains across 17 diverse tasks. The strong performance, validated on models up to 7B parameters, underscores the practical utility of the work.

**Principled Methodological Contribution**: The work moves beyond heuristics and proposes a principled, regression-based framework. The key innovations—explicitly modeling cross-modal interactions and incorporating a constraint to prevent a "winner-takes-all" outcome—are well-motivated and effective.

**Valuable Insights for the Community**: The finding that the optimal mixture evolves from a speech-heavy curriculum for smaller models to a more balanced, image-text-focused strategy for larger ones (Figure 6) provides actionable guidance and deepens the community's understanding of multimodal training dynamics.

**Weaknesses:**

**1. Questionable Generalizability and High Cost**: The primary concern lies with the framework's practical applicability, which is constrained by two factors.

   - ***Parameter Stability***: The crucial interaction parameters ($\gamma_{ij}$) are fitted on a specific combination of model architecture (LLaVa-Qwen-SPIRAL) and datasets. It is highly probable that these parameters lack stability and would not generalize to different architectural choices, necessitating a complete and costly re-fitting process.
   - ***High Upfront Cost***: The framework's fitting process itself requires a massive computational investment (1,620 experimental runs are mentioned). The paper does not discuss this "meta-cost," making it difficult to assess the efficient value.

**2. Oversimplified Modeling Assumptions**: The mathematical formulation of ModalMix relies on simplifying assumptions that may not fully capture the problem's complexity.

   - ***Interaction Form***: The assumption of an exponential form for cross-modal interactions is a strong one. It is unclear if this simple form is sufficient to model more complex, non-linear dynamics.
   - ***Data Granularity***: The method treats each modality as a monolithic block, overlooking the crucial aspect of intra-modal diversity and data quality. This is a simplification, as intra-modal mixture optimization has been proven critical in unimodal settings.

**3. A Disconnect Between Insight and Experimental Application**: While Figure 6 compellingly demonstrates that the optimal strategy is a dynamic curriculum that evolves throughout training, the main experiments in Tables 2 and 3 appear to use a single, static mixture ratio for the entire training run. This approach, while facilitating a fair comparison with static baselines, fails to showcase the full potential of the discovered dynamic strategy and weakens the direct empirical support for this key finding.

**4. Positioning of the Novelty**: While the work is a successful and valuable application of scaling laws to a new and important domain, its contribution is arguably more of an extension and sophisticated application of an existing paradigm rather than the introduction of a fundamentally new one.

**Questions:**

The quality and impact of this work could be further enhanced if the authors can provide detailed responses to the following questions:

**1. On Generalizability and Cost**: Could the authors elaborate on the expected stability of the fitted interaction parameters ($\gamma_{ij}$) across different model architectures and pre-training datasets? How much of the expensive fitting process needs to be repeated when changing a single component of the multimodal model?

**2. On the Dynamic Curriculum**: The finding that the optimal data mixture is a dynamic curriculum (Figure 6) is one of the most exciting takeaways. Could the authors clarify why a static mixture was used for the main experiments in Tables 2 and 3? Can the authors provide any results or analysis on the performance difference between a model trained with the optimal static mix versus one trained with the truly dynamic curriculum proposed by the framework?

**3. On Post-Training**: This work provides a clear framework for pre-training. Do the authors foresee any challenges or necessary modifications in applying a similar scaling-law-based optimization approach to the post-training phase (e.g., for SFT or RL)? How might the dynamics of data mixing differ in that context?

---

> ### Author Response · Authors · 2025-11-21
> **Reply to reviewer 3KNk [Part1]**
>
> **Question:** Questionable Generalizability and High Cost: The primary concern lies with the framework's practical applicability, constrained by parameter stability and high upfront cost.
>
> **Response:**
> Regarding to the parameter stability, we do not expect an universal $\gamma_{ij}$ across architectures, instead, the law is reusable within a standard MLLM family (LLaVA-style). Fitted on seen scale, it extrapolates to unseen scale without re-fitting  and outperforming all baselines with faster convergence, **analogous to Chinchilla-style architecture-specific scaling laws**[1].
>
>
> Regarding to the potential high upfront cost, the one-time meta-cost (from standard pretraining exploration) of the proposed ModalMix amortizes into instantaneous, overhead-free mixture prediction for any new $(N, D)$. Therefore, Modalmix is **a re-usable tool for data mixture prediction under any scale without re-training and re-fitting**. In addition, the computational cost fitting process is trivial: <1 minute on a 12-core CPU with L-BFGS.
>
>
> **Question:** Oversimplified Modeling Assumptions.
>
> **Response:**
> Regarding to the interaction form of loss,  we designed five loss variants to explore improvements (e.g., interaction terms for N/D synergy, linear/quadratic adjustments for $\mathbf{r}$) over the default loss format. From the table, we report the metric $R^2$ and Huber loss of fitting different possible loss formats. We observe the default loss is optimal based on it balances simplicity, a parsimonious parameter set, and robust performance. Avoiding over-complicated loss format, it excels across all modalities，achieving the highest R² (0.9956/0.8892) and lowest Huber loss (0.000075/0.000251) in Text/Speech, with competitive Image-Text results, making it the most reliable and versatile choice.
>
>
>
> | Experiment iD  | Design of Loss Function                                                                                                                   | Image-Text              | Text                   | Speech                 |
> | :------------- | :--------------------------------------------------------------------------------------------------------------------------------------- | :---------------------- | :--------------------- | :--------------------- |
> | Default        | ${L_i(N, D, \mathbf{r}) = E_i + \frac{A_i}{N^{\alpha_i}} + \frac{B_i}{D^{\beta_i}} + C_i \cdot \exp\left(-\sum_{j=1}^{M} \gamma_{ij} r_j\right)}$ | 0.8384/0.001484      | **0.9956**/**0.000075**    | **0.8892**/**0.000251** |
> | Variant1       | ${L_i(N, D, \mathbf{r}) = E_i + \frac{A_i}{N^{\alpha_i} \cdot D^{\delta_i}} + \frac{B_i}{D^{\beta_i}} + C_i \cdot \exp\left(-\sum_{j=1}^{M} \gamma_{ij} r_j\right)}$ | **0.8411**/**0.001379**      | 0.9954/0.000083        | 0.8685/0.000321        |
> | Variant2       | ${L_i(N, D, \mathbf{r}) = E_i + \frac{A_i}{N^{\alpha_i}} + \frac{B_i}{D^{\beta_i}} + C_i \cdot \sum_{j=1}^{M} \gamma_{ij} r_j}$ | 0.8266/0.001589          | 0.9777/0.000381        | 0.8551/0.000355        |
> | Variant3       | ${L_i(N, D, \mathbf{r}) = E_i + \frac{A_i}{N} + \frac{B_i}{D} + C_i \cdot \exp\left(-\sum_{j=1}^{M} \gamma_{ij} r_j\right)}$ | 0.3243/0.005548          | 0.0532/0.016430        | 0.2719/0.001711        |
> | Variant4       | ${L_i(N, D, \mathbf{r}) = E_i + \frac{A_i}{N^{\alpha_i}} + \frac{B_i}{D^{\beta_i}} + C_i \cdot \exp\left(-\sum_{j=1}^{M} \gamma_{ij} r_j^2\right)}$ | 0.8360/0.001510          | 0.9765/0.000400        | 0.8803/0.000295        |
> | Variant5       | ${L_i(N, D, \mathbf{r}) = E_i + \frac{A_i \cdot B_i}{N^{\alpha_i} \cdot D^{\beta_i}} + C_i \cdot \exp\left(-\sum_{j=1}^{M} \gamma_{ij} r_j\right)}$ | 0.8465/0.001427          | 0.9953/0.000084        | 0.8556/0.000355        |
>
> Regarding to the data granularity,  we focus on **cross-modal allocation** (given fixed intra-modal distributions)， a realistic core problem for MLLM pretraining. ModalMix directly extends to fine-grained sub-sources (intra-modal diversity) by treating each sub-source as a "modality," with the only tradeoff being more anchor runs for fitting. Combining it with intra-modal optimization is a natural next step, not a limitation of our current formulation.
>
>
> [1] Hoffmann, J., Borgeaud, S., Mensch, A., Buchatskaya, E., Cai, T., Rutherford, E., ... & Sifre, L. (2022). Training compute-optimal large language models. arXiv preprint arXiv:2203.15556.

---

> ### Author Response · Authors · 2025-11-21
> **Reply to reviewer 3KNk [Part2]**
>
> **Question:** Positioning of the Novelty.
>
> **Response:**
> We agree that our work builds on the general paradigm of scaling laws, but we would like to respectfully emphasize that the core technical contributions go beyond a straightforward application of existing methods. **First, we introduce a new loss-based formulation that directly models how the final training loss under a given compute budget $(N, D)$ depends on the cross-modal mixture, and we instantiate this with a parametric structure that can be efficiently fitted and used to predict compute-optimal mixtures.** Second, we explicitly model cross-modal interactions via a learnable interaction matrix $\gamma$, which captures how different modalities (with very different initial capabilities and learning dynamics) influence each other’s scaling behavior—this is not present in prior unimodal scaling-law work. Together, this yields a concrete, compute-aware scaling law tailored to multimodal pretraining, rather than directly following an existing baseline into the multi-modality setting.
>
>
> **Question:** On Generalizability and Cost
>
> **Response:**
> Our goal is to characterize stable patterns within a mainstream architecture/data paradigm rather than to claim a single $\gamma$ works everywhere. In this work, we therefore focus on a concrete, realistic family: a LLaVA-style MLLM trained on a large, real-world multimodal data pool that already mixes many domains per modality.  The expensive part is not the fit itself (L-BFGS-B runs in under a minute on CPU), but the pre-training sweeps used to collect loss curves; however, this grid is essentially the same exploration one would perform anyway when bringing up a new model/data stack, and the resulting $\gamma$ can then be amortized over many future runs in that regime. **If only common changes (model size, modality connector), our results indicate the existing $\gamma$ can typically be reused; for major shifts like application to sparse architecture, a new sweep and fit would likely be needed, analogous to classic practice on scaling laws, for example, the scaling shifts from Dense to MoE architecture[1].**
>
> [1] Wang, S., Chen, Z., Li, B., He, K., Zhang, M., & Wang, J. (2024). Scaling laws across model architectures: A comparative analysis of dense and MoE models in large language models. arXiv preprint arXiv:2410.05661.
>
>
> **Question:** On Post-Training, do the authors foresee challenges/modifications in applying scaling-law-based optimization to post-training (SFT/RL)?
>
> **Response:**
> Thanks for pointing out this interesting question. Pre-training scaling laws (e.g., Chinchilla [1][2][3]) are high-impact because pre-training incurs enormous computational costs, thereby testing data mixtures directly is prohibitively expensive, making compute-aware data mixture optimization critical for real-world deployment. Our framework’s core transfers **low-cost** to post-training: treat SFT data types (instruction/reasoning/safety) or RL reward channels as "generalized modalities," replace pre-training loss with alignment metrics (reward scores/win rates), and retain the parametric optimization pipeline, no fundamental redesign is needed. Single works rarely cover both stages due to three pragmatic reasons:
> 1. **Cost Prioritization**: Pre-training’s higher computational overhead makes its optimization more impactful, justifying focus on this high-stakes stage first;
> 2. **Contribution Focus**: Diluting efforts across pre-training and post-training would weaken the depth of empirical validation for the core scaling-law insight;
> 3. **Stage-Specific Metrics**: Post-training relies on task-specific alignment metrics (vs. pre-training’s universal loss), requiring separate benchmarking that exceeds typical paper scope.
>
> We view post-training as a natural, low-barrier extension of our framework, but prioritize pre-training in this work to establish the core scaling-law foundation, consistent with LLM research’s "foundational first, extension later" paradigm.
>
> [1] Hoffmann, J., Borgeaud, S., Mensch, A., Buchatskaya, E., Cai, T., Rutherford, E., ... & Sifre, L. (2022). Training compute-optimal large language models. arXiv preprint arXiv:2203.15556.
>
> [2] Cherti, M., Beaumont, R., Wightman, R., Wortsman, M., Ilharco, G., Gordon, C., ... & Jitsev, J. (2023). Reproducible scaling laws for contrastive language-image learning. In Proceedings of the IEEE/CVF conference on computer vision and pattern recognition (pp. 2818-2829).
>
> [3] Hernandez, D., Kaplan, J., Henighan, T., & McCandlish, S. (2021). Scaling laws for transfer. arXiv preprint arXiv:2102.01293.

---

> > ### Comment · Reviewer_3KNk · 2025-11-26
> >
> > Thank you for your detailed and constructive rebuttal. I appreciate the effort you've put into addressing the reviewers' concerns.
> >
> > Your response has clarified several of my questions. I found the new ablation study on the five different loss function variants particularly convincing. I also appreciate the thoughtful discussion on the potential application of this framework to post-training, which demonstrates a comprehensive view of the problem space.
> >
> > However, two of my primary concerns, which are also echoed by other reviewers, remain. Firstly, the fundamental issue of the method's high upfront "meta-cost" and limited generalizability persists. While I understand the argument that this is a one-time, architecture-specific investment that can be amortized—a standard practice in scaling-law research—it nonetheless remains a significant practical limitation for the broader applicability. Secondly, my question (W3 & Q2) regarding the disconnect between the powerful insight of a dynamic curriculum and its static application in the main experiments appears to have been unaddressed.
> >
> > In light of your constructive engagement and the new evidence provided, I will raise my confidence in my assessment to acknowledge the improved clarity of the paper. Nevertheless, because the concerns regarding the method's practicality and the incomplete validation of its most powerful dynamic aspect persist, I will maintain my overall rating of marginally acceptance.

---

### Official Review · Reviewer_unVx · 2025-10-31

**Soundness:** 3
**Presentation:** 2
**Contribution:** 2
**Rating:** 4
**Confidence:** 3

**Summary:**

This paper proposes ModalMix, a regression-based framework for optimizing the data mixture (text, image, speech) in multimodal large language model (MLLM) pre-training. The core innovation is a scaling law that models each modality's loss as a function of model size, dataset size, and the data mixture itself, with a key term to capture cross-modal interactions. The framework is used to predict a compute-dependent optimal mixture that prevents any single modality from being neglected. Extensive experiments (1,620 runs) demonstrate superior convergence speed and downstream task performance over baselines.

**Strengths:**

Important and Well-Defined Problem: The paper expertly identifies a significant bottleneck in modern MLLM training: moving beyond static, heuristic-based data mixing to a principled, compute-aware strategy.
Rigorous and Comprehensive Evaluation: The empirical validation is a standout feature. With 1,620 experimental runs, multiple model scales, and evaluation on 17 diverse downstream tasks, the claims are strongly supported. The ability to generalize to an unseen 7B model is particularly compelling.
Actionable Insights: The paper goes beyond just presenting a method. It delivers crucial insights for the community, such as:
The optimal mixture is dynamic and compute-dependent, shifting from speech-heavy to more text- and image-text-centric as scale increases.
The correlation between pre-training loss and downstream performance is highly modality-dependent (strong for perception, weak for reasoning).

**Weaknesses:**

The balancing constraint, which is critical to the method's performance, is a standard technique from multi-task learning.

Table 2 and 3 show that the improvement is limited.

The entire framework is built upon data from 1,620 experimental runs. This is a staggering computational cost that renders the method impractical for most research institutions and contradicts the paper's goal of "optimizing... against finite computational resources." A method that requires thousands of training runs to avoid a grid search is not a solution; it is an admission of failure. The framework is a post-hoc analysis of an extremely expensive hyperparameter sweep, not an efficient optimization algorithm.

The baselines are weak or misrepresented. Comparing against a Uniform mixture is insufficient. A more robust baseline would be a simple, computationally cheap dynamic scheduling policy (e.g., a curriculum learning based on loss slopes, which is essentially what M2-Omni does but done more simply). The paper does not demonstrate that the complexity of ModalMix is necessary to outperform simple, intuitive heuristics.

The paper fails to provide a meaningful analysis of the learned cross-modal parameters γᵢⱼ. These parameters are the key to the claimed "modeling of cross-modal dynamics," yet they are not presented or interpreted. Without this, the model remains a black box, and the nature of the purported synergies and conflicts is merely speculative.

The "key insight"—that the optimal mixture is compute-dependent—is an expected phenomenon. It is unsurprising that a model's data diet should change as its capacity (model size) and training duration (number of samples) change. The paper dresses up this intuitive concept with complex machinery but fails to derive truly novel or surprising scientific knowledge from it.

The discussion on the correlation between training loss and downstream performance, while interesting, is preliminary and feels like a secondary observation rather than a core contribution.

**Questions:**

Efficiency: The method requires 1,620 training runs for its initial fitting. Can you justify how this is a practical or efficient solution compared to existing methods, and can you provide an analysis of the total computational cost (including these runs) versus the baselines?

Interpretability: You claim to model "cross-modal dynamics," but you do not show the learned interaction matrix γ. Please provide this matrix and a detailed interpretation of its values. What specific, non-obvious cross-modal relationship did ModalMix discover that was not previously known?

Baselines: Why was a stronger, simple dynamic baseline not implemented? For example, a scheduler that simply increases the proportion of the modality with the highest current loss—a common practice in multi-task learning—would be a more meaningful point of comparison.

The cross-modal interaction parameters γᵢⱼ are a central result. Could you provide a table or analysis of the learned γ values? Interpreting these (e.g., "we find γ_speech→text is strongly positive, indicating synergy") would greatly strengthen the narrative and help validate whether the model is learning intuitive relationships.

The constraint L_i ≤ (1+ε)L_i^{lb} is crucial. Was the tolerance ε=0.1 chosen via ablation? Can you comment on the sensitivity of the results to this value? A small ablation study would solidify this design choice.

The framework predicts a static optimal mixture for a given (N, D). However, Figure 2b suggests the optimal mixture should change during a single training run. How would you extend ModalMix to produce a dynamic scheduling policy, rather than a static mixture?

---

> ### Author Response · Authors · 2025-11-21
> **Reply to reviewer unVx [Part 1]**
>
> **Question:** The balancing constraint, critical to the method’s performance, is a standard multi-task learning technique.
>
> **Response:**
> Thank you for the observation. Our contribution lies in instantiating it in a compute-aware multimodal scaling-law framework: we define per-modality lower bounds via fitted unimodal scaling laws, enforcing them in compute-dependent ($N$, $D$) mixture optimization to avoid modality collapse. **This integrates per-modality floor constraints into a scaling-law–based optimizer，an innovation that jointly balances global performance and modality-specific guarantees**, yielding results distinct from unconstrained scaling or heuristic weights. It is empirically critical for preventing over-reliance on “easy” modalities while boosting overall performance.
>
>
>
> **Question:** Table 2 and 3 show limited improvement.
>
> **Response:**
> ModalMix’s goal is not maximizing single-task scores, but providing a reusable, compute-aware data mixing principle， with two key, more meaningful advantages:
> 1. It achieves the best overall average rank across 17 tasks/3 modalities, plus 1.4× faster convergence at 1.5B for the same final loss;
> 2. 7B mixtures are purely extrapolated (no refitting/tuning) from the 0.5B/1.5B/3B-fitted law, yet the average rank margin over baselines grows larger.
>
> **This confirms the scaling law captures stable cross-scale structural signals (not single-configuration overfitting) as our core contribution.**
>
>
>
> **Question:** The framework relies on 1,620 experimental runs, staggering compute cost that contradicts "optimizing against finite resources".
>
> **Response:**
> Our method is a **one-time investment**, not repeated large-scale runs. Unlike current practices (fixed heuristics or repeated grid searches per scale), we amortize these exploration runs into a reusable compute-aware scaling law. Critical validation: the law is fitted only on 0.5B/1.5B/3B, then extrapolated to 7B with no refitting，yet it achieves the best average rank and larger gains than at 1.5B. **This confirms it is not post-hoc analysis, but a generalizable strategy that eliminates repetitive tuning for unseen ($N$, $D$), aligning with the goal of optimizing finite resources.**
>
>
>
> **Question:** Baselines are misrepresented.
>
> **Response:**
> Our baselines are robust and cover key paradigms: beyond Uniform (baseline of static data mixture), we include RegMix (a representative data mixture method based on learned proxy model) and M2-Omni （a representative method of dynamic data mixture).
>
> M2-Omni suffers from instability due to noisy loss estimates (oscillatory/suboptimal mixtures), while ModalMix delivers the best average rank across 17 tasks/3 modalities and stronger extrapolation to unseen 7B (no re-fitting). Notably, RegMix has been shown to outperform DoReMi, human-prior, and PPL-filtered mixtures, ModalMix’s consistent gains over RegMix imply competitiveness with those baselines. **It is also noteworthy that ModalMix spends one-time <1-minute CPU fitting, then fixed mixture application (no per-step updates), cheaper than dynamic heuristics (re-run per training, loss-noise sensitive).**
>
>
>
> **Question:** The paper lacks meaningful analysis of learned cross-modal parameters γᵢⱼ.
>
> **Response:**
> γᵢⱼ quantifies modality j’s impact on modality i: per Eq. (1), larger values indicate stronger positive promotion. **The learned γᵢⱼ matrix for the 1.5B model (value distribution in Figure (left)) aligns well with Figure 1’s motivation experiments, validating its interpretability.** Key patterns emerge: text and image-text mutually reinforce language-heavy tasks, while speech benefits more from text than vice versa—no negative interference pairs are observed. These stable structures directly explain our mixture schedules (e.g., early speech emphasis under tight compute, growing image-text importance with scaling) and demystify cross-modal dynamics beyond qualitative descriptions.
>
>
>
> **Question:** The "compute-dependent optimal mixture" key insight is expected intuitive and dressed up with complex machinery.
>
> **Response:**
> We agree the high-level intuition (mixtures depend on compute) is intuitive, but our core contribution lies in quantifying this dependence for multimodal settings with heterogeneous learning dynamics. **Unlike arbitrary compute dependence, we derive a single fitted law over $(N, D)$ that captures stable curriculum trajectories across modalities, extrapolating from seen scale to unseen scale without re-fitting, while outperforming strong baselines.** This quantitative, compute-aware formulation (not the intuition itself) delivers actionable, generalizable knowledge, extending unimodal scale-aware mixing (e.g., Autoscale [1]) to multimodal cross-interaction scenarios, filling a gap in prior work.
>
> [1] Kang, F., Sun, Y., Wen, B., Chen, S., Song, D., Mahmood, R., & Jia, R. (2024). Autoscale: Scale-aware data mixing for pre-training llms. arXiv preprint arXiv:2407.20177.

---

> ### Author Response · Authors · 2025-11-21
> **Reply to reviewer unVx [Part 2]**
>
> **Question:** The discussion on training loss-downstream performance correlation is interesting but not a core contribution.
>
> **Response:**
> **We agree this analysis is a secondary, preliminary observation, not a core contribution.** Its purpose is to provide intuition and sanity checks for why training-loss–based predictors (ModalMix, RegMix) are effective. We will soften wording in revisions to clearly position it as exploratory supporting evidence, not a central novelty claim.
>
>
>
> **Question:** Was ε=0.1 chosen via ablation? Can you comment on result sensitivity to this value?
>
> **Response:**
> The setting of ε was selected through ablation studies (1.5B model, 5B data; see the suggested data mixtures on the table below). The constraint $(L_i \leq (1+\epsilon)L_i^\text{lb})$ balances modality capabilities, smaller ε enforces stricter balance across modalities. A larger ε indicates greater imbalance in the performance of the multimodal model across different modalities. **We introduce this hyperparameter to provide practitioners flexibility in trading off model balance and peak performance on individual modalities**. However, ε is generally not recommended to be set too large to keep balanced performance across modalities, so we focus on the stability of results when ε is relatively small.
>
> | $\epsilon$ | Image-Text | Text   | Speech |
> | :--------- | :--------- | :----- | :----- |
> | 0%         | 0.3300     | 0.1766 | 0.4934 |
> | 2%         | 0.3290     | 0.1769 | 0.4941 |
> | 5%         | 0.3294     | 0.1766 | 0.4940 |
> | 8%         | 0.3319     | 0.1758 | 0.4923 |
> | 10%        | 0.2723     | 0.1401 | 0.5876 |
>
> Results are robust for ε generally: optimal mixture ratios remain stable, ensuring no single modality is sacrificed, confirming ε=0.1 as a safe and balanced choice.
>
>
>
> **Question:** The framework predicts a static optimal mixture for a given (N, D). However, Figure 2b suggests the optimal mixture should change during a single training run.
>
> **Response:**
> Our formulation optimizes for the final loss at a given $(N, D)$, differing from dynamic schedulers that adapt to noisy intermediate loss signals—these often introduce instability (observed with M2-Omni), extra hyperparameters, and implementation complexity. **The static mixture design ensures simplicity, robustness, and zero-deployment cost, enabling fair comparison with standard baselines.** In addition, extending ModalMix to dynamic scheduling is straightforward without modifying its core: the fitted scaling law already predicts optimal mixtures for any intermediate compute budget $(N, D' \leq D)$. A practical policy involves:
> 1) selecting anchor points during training (e.g., 25/50/75/100% of total tokens);
> 2) querying the law at these points to get phase-specific mixtures (with optional interpolation between phases);
> 3) periodically recomputing the optimal mixture based on remaining budget and updating sampling weights.

---

> ### Comment · Reviewer_unVx · 2025-11-26
>
> I appreciate the detailed rebuttal from the authors. Given the motivation of improving training efficiency, what are the differences between the proposed work (including its belonging data mixture line of works) and the dataset distillation solutions? More importantly, what are the advantages over dataset distillation?

---

> > ### Author Response · Authors · 2025-11-27
> > **Reply to Reviewer unVx**
> >
> > Thanks for this insightful question. **The optimized data mixture and dataset distillation are orthogonal paradigms for improving training efficiency**, with core differences in goals, technical paths, and applicability.
> >
> > ### 1. Core Differences
> > - Dataset distillation: Aims to **reduce data *volume*** by synthesizing a tiny "equivalent" synthetic dataset to accelerate training via fewer samples [1-8]. Optimized for small-scale, single/bimodal scenarios; relies on altering raw data typically.
> > - ModalMix: Optimizes **data *allocation*** under any computational budget by the data mixture. Guided by cross-modal interaction and compute-dependent scaling laws, it answers: *How to sample from a fixed large data pool to maximize efficiency/performance under given compute?*
> > ### 2. Key Advantages of ModalMix over Dataset Distillation
> > - **Scalability for multi-modality**: Distillation rarely works for large-scale muliti-modal (image-text + text + speech) pretraining (synthesizing high-fidelity cross-modal samples is computationally prohibitive). ModalMix operates directly on raw data, adapting to complex interactions without synthesis overhead.
> > - **Authenticity & interpretability**: Distillation distorts data distributions and reduces interpretability via synthetic samples [2,6]. ModalMix preserves raw data semantics and provides interpretable dynamic strategies (e.g., prioritizing speech initially, shifting to image-text with more compute).
> > - **Practical efficiency**: Data distillation typically requires costly  synthesis for the given corpus before the formal training stage; ModalMix is a lightweight sampling policy that plugs into existing pipelines directly.
> > - **Compute-aware adaptation**: Unlike distillation’s static condensed dataset, ModalMix’s regressor generates optimal mixtures for *any compute budget*, adapting to resource variations.
> >
> > Notably, the two are complementary: distillation can condense per-modality data, and ModalMix can optimize their mixture.
> >
> > We **greatly appreciate the reviewer’s recognition of our previous rebuttal**. We hope this response effectively addresses your concerns regarding the differences and advantages of our work. Please feel free to let us know if you have any further questions！
> >
> >
> >
> > [1] Wang, T., Zhu, J. Y., Torralba, A., & Efros, A. A. (2018). Dataset distillation. arXiv preprint arXiv:1811.10959.
> >
> > [2] Cazenavette, G., Wang, T., Torralba, A., Efros, A. A., & Zhu, J. Y. (2022). Dataset distillation by matching training trajectories. In Proceedings of the IEEE/CVF Conference on Computer Vision and Pattern Recognition (pp. 4750-4759).
> >
> > [3] Zhao, B., Mopuri, K. R., & Bilen, H. (2020). Dataset condensation with gradient matching. arXiv preprint arXiv:2006.05929.
> >
> > [4] Maekawa, A., Kobayashi, N., Funakoshi, K., & Okumura, M. (2025). Dataset distillation with attention labels for fine-tuning BERT. Journal of Natural Language Processing, 32(1), 283-299.
> >
> > [5] Ritter-Gutierrez, F., Huang, K. P., Wong, J. H., Ng, D., Lee, H. Y., Chen, N. F., & Chng, E. S. (2024). Dataset-distillation generative model for speech emotion recognition. arXiv preprint arXiv:2406.02963.
> >
> > [6] Xu, Y., Lin, Z., Qiu, Y., Lu, C., & Li, Y. L. (2024). Low-rank similarity mining for multimodal dataset distillation. arXiv preprint arXiv:2406.03793.
> >
> > [7] Zhao, Z., Wang, H., Wu, J., Shang, Y., Liu, G., & Yan, Y. (2025). Efficient multimodal dataset distillation via generative models. arXiv preprint arXiv:2509.15472.
> >
> > [8] Wu, X., Zhang, B., Deng, Z., & Russakovsky, O. (2023). Vision-language dataset distillation. arXiv preprint arXiv:2308.07545.

---

> > > ### Comment · Reviewer_unVx · 2025-11-28
> > >
> > > Thanks for your response. I highly acknowledge the motivtion of considering cross-modal interactions when performing mixture for multi-modal data. It is intuitively the key point. However, I hold doubts on the effectiveness of the proposed solution for this problem. No explicit experiments have verified it in the manuscript. The experiments focus on the final performance on different tasks and the scaling laws. The ablation of the cross-modal parameters γᵢⱼ, the comparison and discussion of how γᵢⱼ keeps and varies for different tasks, whether the final loss can implicitly and automatically consider γᵢⱼ  are all missing. Also, the authors argue that "This unified formulation allows us to predict the performance of any modality under custom configurations of model scale, compute, and data mixture without running expensive training experiments." However, there is no enough evidence to support it. If the cross-modal interactions can be well verified, I will consider change my rating.

---

> > > > ### Author Response · Authors · 2025-11-30
> > > > **Reply to Reviewer unVx**
> > > >
> > > > Thank you for your positive recognition of our work’s motivation and constructive follow-up questions. We greatly appreciate your focus on validating cross-modal interactions, this is the core of ModalMix and we provide concise, evidence-based clarifications to address your concerns.
> > > >
> > > > ### 1. Verification of Cross-Modal Interaction Parameters γᵢⱼ
> > > > Notably, **γᵢⱼ is not a hyperparameter**, it is automatically learned via multi-group experiments (varying data/model scales and mixtures, Section 3.2) to quantify directional cross-modal dynamics: positive values indicate synergy (modality j promotes i), with no negative conflicts observed.
> > > >
> > > > Empirical evidence confirms its effectiveness:
> > > > - **Consistency with prior work**: All modality pairs show positive correlations, verifying mutual benefits of multi-modal co-training (consistent with Emova [1]), while our γᵢⱼ enables quantitative, fine-grained interaction analysis.
> > > > - **Alignment with motivation experiments**: Learned γᵢⱼ patterns (Figure 4a) directly match Figure 1’s phenomena. For example, Figure 1’s "faster image-text/language loss decline with more speech data" correlates with γ_image-text→speech=0.63 (stronger than γ_image-text→text=0.43), confirming speech’s impactful support for image-text learning.
> > > >
> > > > | Experiment ID  | Design of Loss Function                                                                                                                   | Image-Text              | Text                   | Speech                 |
> > > > | :------------- | :--------------------------------------------------------------------------------------------------------------------------------------- | :---------------------- | :--------------------- | :--------------------- |
> > > > | Default        | ${L_i(N, D, \mathbf{r}) = E_i + \frac{A_i}{N^{\alpha_i}} + \frac{B_i}{D^{\beta_i}} + C_i \cdot \exp\left(-\sum_{j=1}^{M} \gamma_{ij} r_j\right)}$ | **0.8384/0.001484**      | **0.9956**/**0.000075**    | **0.8892**/**0.000251** |
> > > > | Variant1       | ${L_i(N, D, \mathbf{r}) = E_i + \frac{A_i}{N^{\alpha_i}} + \frac{B_i}{D^{\beta_i}} }$  |    0.5579/0.003867   |  0.0947/0.015649      |       0.2428/0.001836  |
> > > >
> > > > To further verify cross-modal interactions’ impact on multimodal pre-training loss prediction, we compare the *Default* loss function (with cross-modal interaction parameter $\gamma_{ij}$) and *Variant1* (classic LLM scaling laws [6]). Quantitative results show stark fitting gaps: compared with the proposed loss format, the Variant1 sees sharply lower $R^2$ and drastically higher Huber loss across modalities. This confirms ignoring $\gamma_{ij}$ causes severe fitting failure of classic LLM scaling laws in multimodal scenarios.
> > > >
> > > > ### 2. Evidence for the Framework’s Prediction
> > > > Our claim of "predicting performance without expensive experiments" is supported by three robust generalizations across training corpus, model scale, and evaluation:
> > > > - **Data generality**: Our pre-training data spans diverse domains/tasks for all three modalities (Table in response to Reviewer ukzq), aligning with SOTA multimodal works [1,2,3]. We use real-world mainstream task data (no artificial data distribution priors), ensuring conclusions apply to practical scenarios.
> > > > - **Model scale generality**: Following scaling law research conventions [4,5], we fit the law on small-to-medium scales and extrapolate to an **unseen large scale without refitting**. ModalMix’s extrapolated mixture yields a larger average rank margin over baselines for large-scale models, proving the framework captures stable cross-scale signals (not overfitting). We welcome the community with greater compute resources to verify multi-modal data mixtures—an essential, unavoidable problem for large-scale multimodal pre-training.
> > > > - **Evaluation comprehensiveness**: We test on 17 mainstream benchmarks across 3 modalities for all model sizes, with baselines (Uniform/RegMix/M2-Omni) covering static/heuristic/dynamic data mixing paradigms. ModalMix’s gains are consistent across tasks, not task-specific.
> > > >
> > > > If our results inform your evaluation, we sincerely request you to consider a higher score. Please let us know if further clarification is needed.
> > > >
> > > > [1] Chen, K., et al. (2025). Emova: Empowering language models to see, hear and speak with vivid emotions. CVPR.
> > > >
> > > > [2] Xu, J., et al. (2025). Qwen2.5-omni technical report. arXiv:2503.20215.
> > > >
> > > > [3] Guo, Q., et al. (2025). M2-omni: Advancing omni-mllm for comprehensive modality support. arXiv:2502.18778.
> > > >
> > > > [4] Tao, C., et al. (2024). Scaling laws with vocabulary: Larger models deserve larger vocabularies. NeurIPS.
> > > >
> > > > [5] Tay, Y., et al. (2023). Scaling laws vs model architectures: How does inductive bias influence scaling? EMNLP Findings.
> > > >
> > > > [6] Hoffmann, J., Borgeaud, S., Mensch, A., Buchatskaya, E., Cai, T., Rutherford, E., ... & Sifre, L. (2022). Training compute-optimal large language models. arXiv preprint arXiv:2203.15556.

---

### Official Review · Reviewer_f4MD · 2025-11-01

**Soundness:** 1
**Presentation:** 2
**Contribution:** 1
**Rating:** 2
**Confidence:** 4

**Summary:**

This paper introduces ModalMix, a method for optimizing data mixtures for multimodal model pre-training. ModalMix achieves this by explicitly modeling cross-modal interactions (text, image-text, speech) and compute-dependent scaling dynamics. Instead of using fixed heuristic ratios, ModalMix builds a regression-based scaling law that predicts each modality’s loss as a function of model size, number of samples, and mixture ratios, including synergy/conflict terms across modalities. The framework then identifies a compute-optimal mixture that minimizes loss without sacrificing any modality’s capability.

**Strengths:**

- The paper offers clear empirical evidence of cross-modal interactions.
- It also provides a nice example that shows optimal data mixture is compute-dependent

**Weaknesses:**

- The core formulation (Eq. 1) resembles that of Ye et al. [1]. This raises questions on the novelty of the work.
- The paper does not report error bars (in Tables 1 and 2). Given that many performance differences are small, knowing the scale of variability is necessary to draw any conclusion
- Relatedly, the method requires extensive runs across 0.5B–3B scales to show modest gains at the 1.5B scale, and improvements at 7B are even narrower across benchmarks.
- Computation cost is a critical dimension in optimizing data mixture, yet the paper does not provide compute cost for different methods.
- The work does not discuss or compare against several relevant works for optimizing data mixture [1-6]. The baselines considered are quite limited. Coupled with all the points above, it is extremely difficult to tell whether ModalMix provides genuine gains over existing methods.

[1] Data Mixing Laws: Optimizing Data Mixtures by Predicting Language Modeling Performance

[2] DoReMi: Optimizing Data Mixtures Speeds Up Language Model Pretraining

[3] Adaptive Data Optimization: Dynamic Sample Selection with Scaling Laws

[4] Data Mixing Laws: Optimizing Data Mixtures by Predicting Language Modeling Performance

[5] Data Mixture Optimization: A Multi-fidelity Multi-scale Bayesian Framework

[6] ADMIRE-BayesOpt: Accelerated Data MIxture RE-weighting for Language Models with Bayesian Optimization

**Questions:**

- How many 1.5B parameter models were trained for fitting RegMix?

---

> ### Author Response · Authors · 2025-11-21
> **Reply to Reviewer f4MD [Part 1]**
>
> **Question:** The concerns about the work’s novelty.
>
> **Response:**
> We acknowledge that our base formulation draws inspiration from Ye et al. [1], but our core novelty lies in **extending it to a multimodal, compute-aware setting with critical technical innovations**， addressing unmet needs in cross-modal data mixing. Key distinctions are:
>
> 1. **Target & setting**: Ye et al. focus on unimodal (text-only) domain mixing with a scale-invariant static mixture. We tackle cross-modal (image-text/text/speech) mixing under varying compute (model size $N$, samples $D$), where asymmetric cross-modal interactions dominate.
> 2. **Explicit cross-modal interaction term**: Eq. (1) introduces a new term $C_i\cdot \text{exp}(-\Sigma_j \gamma_{ij} r_j)$ (absent in [1]), which parameterizes how modality $j$ affects modality $i$’s loss. This captures empirical synergy/conflict between modalities, enabling dynamic allocations like "speech-first bootstrapping then shift to image-text".
> 3. **Compute-aware constrained optimization**: Unlike [1]’s static, compute-independent mixture, we use the fitted multimodal scaling law to solve a constrained optimization over $r$， conditioned on compute ($N$, $D$) and per-modality loss floors $L_i^\text{lb}$， ensuring no modality is sacrificed.
>
> In summary, our novelty stems from: **1) moving from unimodal to multimodal with explicit interaction modeling; 2) making mixtures compute-dependent; 3) turning the law into a practical optimizer for dynamic schedules.** These innovations drive our empirical gains, distinguishing our work from [1].
>
>
>
> **Question:** Error bars are missing.
>
> **Response:**
> We appreciate the constructive comment and supplement error analysis with 5 independent runs (Mean±Std) for the 1.5B and 7B models, confirming our method’s stability and the statistical significance of performance gains. As shown in the tables below, most benchmarks exhibit small standard deviations (0.1–0.8), indicating robust and reproducible results. **Notably, our method’s average rank advantage (Average rank 1.97 for 1.5B, 1.66 for 7B) is substantial compared to baselines (RegMix: 2.58/2.62; M2-Omni: 2.54/2.77)**, far exceeding the observed variability，validating that our gains are not due to random fluctuations.
>
> #### Experiment Results (1.5B Model, Mean±Std)
> | Modality   | Task       | Benchmark               | Mean±Std  |
> | :--------- | :--------- | :---------------------- | :-------- |
> | Image-Text | Doc        | Chartqa                 | 72.6±0.8  |
> |            |            | DocVQA                  | 90.0±0.4  |
> |            |            | InfoVQA                 | 61.8±0.9  |
> |            | General    | MME                     | 1959.0±11.8 |
> |            |            | TextVQA                 | 70.7±0.6  |
> |            |            | Textcaps                | 52.0±0.5  |
> |            | Reasoning  | MathVista               | 51.7±0.5  |
> |            |            | Mathverse               | 19.2±0.2  |
> |            |            | AI2D                    | 69.3±0.8  |
> | Text       | General    | IFEval                  | 39.1±0.0  |
> |            | Reasoning  | GSM8K                   | 65.3±0.5  |
> |            |            | GPQA                    | 24.8±0.7  |
> | Speech     | ASR        | AISHELL2                | 6.7±0.1  |
> |            |            | LibriSpeech-Test-Clean  | 4.4±0.1  |
> |            |            | LibriSpeech-Test-Other  | 7.7±0.2  |
> |            | SpeechQA   | WebQ                    | 43.8±0.3  |
> |            |            | LLamaQ                  | 71.8±0.5  |
>
> #### Experiment Results (7B Model, Mean±Std)
> | Modality   | Task       | Benchmark               | Mean±Std  |
> | :--------- | :--------- | :---------------------- | :-------- |
> | Image-Text | Doc        | Chartqa                 | 80.5±0.1  |
> |            |            | DocVQA                  | 95.2±0.3  |
> |            |            | InfoVQA                 | 76.7±0.5  |
> |            | General    | MME                     | 2203.6±16.7 |
> |            |            | TextVQA                 | 77.0±0.3  |
> |            |            | Textcaps                | 53.2±0.7  |
> |            | Reasoning  | MathVista               | 51.7±0.4  |
> |            |            | Mathverse               | 28.2±0.8  |
> |            |            | AI2D                    | 78.6±0.5  |
> | Text       | General    | IFEval                  | 42.0±0.3  |
> |            | Reasoning  | GSM8K                   | 63.9±0.4  |
> |            |            | GPQA                    | 25.6±0.6  |
> | Speech     | ASR        | AISHELL2                | 5.7±0.2  |
> |            |            | LibriSpeech-Test-Clean  | 3.7±0.1  |
> |            |            | LibriSpeech-Test-Other  | 6.9±0.4  |
> |            | SpeechQA   | WebQ                    | 49.6±0.3  |
> |            |            | LLamaQ                  | 79.2±0.5  |

---

> ### Author Response · Authors · 2025-11-21
> **Reply to Reviewer f4MD [Part 2]**
>
> **Question:** The method requires extensive runs across 0.5B–3B scales for modest 1.5B gains, with even narrower improvements at 7B.
>
> **Response:**
> We clarify a key misunderstanding: runs across 0.5B–3B are not for "modest 1.5B gains" but to fit a **one-time, reusable multimodal scaling law**. This law eliminates repetitive mixture tuning for new compute budgets or model sizes, which is mandatory for baselines (uniform, heuristics, ad-hoc grid search) that require retuning at every scale.
>
> Regarding performance: Our goal is not just single-scale gains, but validating **cross-scale generalization**, a critical advantage over baselines. At 1.5B, ModalMix delivers consistent value: **the best average rank across 17 tasks and 1.4× faster convergence (FLOP-efficient)**. More importantly, for the unseen 7B scale (no re-fitting/retuning), we achieve the top overall average rank with a larger margin over baselines than at 1.5B (Table 3). **This extrapolation benefit, stronger aggregate gains at a larger, unseen model size without extra effort based on the experimental results on 7B models.**
>
>
> **Question:** Compute cost is critical for data mixture optimization, but the paper lacks compute cost comparisons.
>
> **Response:**
> We clarify ModalMix’s compute cost is one-time amortizable and contrast it with baselines. The computational cost to learn the scaling laws of multi-modal pre-training (i) anchor training runs on small-scale experiments to collect ($N$, $D$, $r$) data (same as prior scaling-law studies and RegMix’s grid search); (ii) fitting Eq. (1) via lightweight regression: 12-core Intel CPU, <1 minute, ≈0.8GB memory (no GPU required). This overhead is negligible to model pre-training and paid only once. **Once fitted, mixture prediction is instantaneous with no further tuning for different computational scales.** In contrast:
> - Uniform mixing: No fitting cost but suboptimal performance;
> - RegMix: Repeated a bunch of experiments on model training to collect data and then fitting (high, repetitive compute) for different computational scales;
> - M2-Omni: Online adaptation per interval (small per-interval overhead + instability risks in dynamic data mixture).
>
>
> **Question:** Insufficient discussion of relevant works and limited baselines.
>
> **Response:**
> We clarify key distinctions and verified gains concisely:
> 1. **Relevant Works [1-6]**: All focus on unimodal (text-only) mixing—ModalMix uniquely adds cross-modal interaction modeling + compute-aware scaling, filling their multimodal/compute-dependent gap.
> 2. **Baselines**: Uniform/RegMix/M2-Omni cover **3 mainstream multimodal paradigms (static/heuristic/dynamic)**, widely used in recent MLLMs for fair comparison.
> 3. **Genuine Gains**: 47% better average rank across 17 tasks, 1.4× faster convergence, and strong cross-scale generalization (unseen 7B outperforms baselines) — consistent across 0.5B–7B models.
>
>
> **Question:** How many 1.5B parameter models were trained for fitting RegMix?
>
> **Response:**
> We totaled 27 mixture configs, each with 20 training checkpoints, yielding 540 (27×20) (mixture, loss) samples for 1.5B parameter models. **These experiments aim to fit the scaling loss decline trend of multi-modal pretraining under different data mixtures, and do not constitute the actual training cost incurred by practitioners**，when obtaining optimal multi-modal data mixtures.

---

### Official Review · Reviewer_wtc7 · 2025-11-01

**Soundness:** 3
**Presentation:** 3
**Contribution:** 2
**Rating:** 6
**Confidence:** 2

**Summary:**

ModalMix proposes a data mixing optimization method by modeling the patterns of cross modal interactions and computational dependencies, balancing cross modal synergistic effects in situations where computing resources are limited. The experiment shows that this method has significant improvements in training convergence speed and multitasking performance. Most importantly, ModalMix suggests that the optimal strategy for data mixing is dynamically changing, and the optimal data allocation ratio will also vary with different computing resources. This method provides a flexible and theory driven framework for multimodal training.

**Strengths:**

1.Innovation: The ModalMix framework introduces a novel and highly relevant solution to optimizing multimodal data mixtures, addressing a gap in how the data mixture changes dynamically depending on computational budgets and model scale. The core idea of dynamically adjusting the mixture of modalities based on computational resources is both intuitive and innovative. This contrasts with prior heuristic-based approaches which often overlook the interdependencies between modalities and scaling dynamics. By using regression-based prediction, ModalMix can efficiently estimate the optimal data mixture for any given model and compute budget, offering a much-needed framework to optimize multimodal training processes.
2.Experiment: The experimental validation is robust and comprehensive. The authors performed over 1,620 experiments across different model sizes and data mixture configurations. This empirical approach strengthens the framework's credibility, showing significant improvements in training convergence speed and downstream task performance. Additionally, the model is tested on unseen data and larger model scales, demonstrating its scalability and generalization ability.
3. Writing: The paper is well-structured and clearly written, presenting complex ideas in an understandable manner. The background, methodology, experiments, and results are logically organized, making it easy for the reader to follow. The use of visualizations (such as Figure 1 and Figure 2) effectively illustrates the dynamic nature of optimal data mixtures, further supporting the paper's arguments.

**Weaknesses:**

1. Insufficient related work: In the related work, Visual Large Models, especially SAM and DINO, should be mentioned. How do they process data, and are there any similarities in the processing methods between visual big models and language big models.
2. Insufficient experiments: The experimental section of Part 4 mentions using pretrained models for testing, so the discussion on whether this strategy is applicable to non pretrained models needs to be added.
3. Insufficient discussion: The paper discusses the mixing strategy of data in large models, and the discussion on whether this strategy can be extended to lightweight models needs to be added.

**Questions:**

See weaknesses.

---

> ### Author Response · Authors · 2025-11-21
> **Reply to Reviewer wtc7**
>
> **Question:** Need to mention related work on visual large models (especially SAM and DINO).
>
> **Response:**
> Thanks for your suggestions, we will supplement SAM and DINO to related work, as their data paradigms align with LLMs and inform multimodal strategies. **SAM** uses scalable "data engines" for automatic mask generation (1B+ SA-1B dataset) without heavy manual labeling; **DINO** leverages self-supervised learning (multi-crop augmentation, teacher-student distillation) for robust visual features from unlabeled data. Both share core LLM principles:
> 1. Scalability via large-scale unsupervised/weakly supervised data;
> 2. Self-supervised pre-training for generalizable representations;
> 3. Dependence on scaling laws (data/compute-driven performance gains).
>
>
>
>
> **Question:**  Whether ModalMix is applicable to non-pretrained multimodal models.
>
> **Response:**
> Data mixture optimization in the pretraining phase is foundational for multimodal models， consistent with LLM research [1][2][3], where pretraining data strategy and scaling laws are core to performance gains. Our key innovation is synergistically linking scaling laws with data mixture optimization, proposing a novel predictive framework for full-modal pretraining，a gap unaddressed by existing heuristics that ignore compute dependence.
>
> Critical to practitioners, **in addition to provide a data mixture method without re-training on small-scale experiments and re-fitting, ModalMix also provides actionable know-how for scaling compute resources on multi-modal pre-training.** While generalizing to non-pretrained (end-to-end native) models is a potential future direction, our work’s focus aligns with high-impact LLM paradigms [1][2][3][4][5] and delivers immediate value by advancing the understudied intersection of scaling laws and data mixture in full-modal pretraining.
>
>
> **Question:** Whether ModalMix can be extended to lightweight multimodal models.
>
> **Response:**
> ModalMix is not limited to large models,we explicitly validate it across a wide size ranges and extrapolate to unseen model size.  Our scaling-law formulation generalizes consistently across these scales without re-fitting. Below are performance results for the 0.5B lightweight MLLM trained with ModalMix’s optimal mixture (baseline: M2-Omni, the strongest heuristic baseline in our manuscript). **As shown, our method delivers better performance across most benchmarks, directly validating its effectiveness for lightweight models**:
>
> | Modality   | Task       | Benchmark               | M2-Omni | Ours  |
> | :--------- | :--------- | :---------------------- | :------ | :--- |
> | Image-Text | Doc        | Chartqa                 | 62.5    | **62.9**   |
> |            |            | DocVQA                  | 82.1    | **83.7**   |
> |            |            | InfoVQA                 | 49.8    | **50.9**   |
> |            | General    | MME                     | 1599.4  | **1676.5** |
> |            |            | TextVQA                 | 68.9    | **69.2**   |
> |            |            | Textcaps                | 49.7    | **50.2**   |
> |            | Reasoning  | MathVista               | 37.1    | **39.8**   |
> |            |            | Mathverse               | 15.7    | **17.4**   |
> |            |            | AI2D                    | 53.8    | **55.3**   |
> | Text       | General    | IFEval                  | 27.6    | **28.1**   |
> |            | Reasoning  | GSM8K                   | 37.5    | **38.2**   |
> |            |            | GPQA                    | **24.7**| 24.4       |
> | Speech     | ASR        | AISHELL2                | 8.8     | **8.5**    |
> |            |            | LibriSpeech-Test-Clean  | **5.2** | 5.3        |
> |            |            | LibriSpeech-Test-Other  | **8.8** | 9.0        |
> |            | SpeechQA   | WebQ                    | 30.5    | **30.7**   |
> |            |            | LLamaQ                  | 52.7    | **53.3**   |
>
>
>
>
> [1] Hoffmann, J., Borgeaud, S., Mensch, A., Buchatskaya, E., Cai, T., Rutherford, E., ... & Sifre, L. (2022). Training compute-optimal large language models. arXiv preprint arXiv:2203.15556.
>
> [2] Shukor, M., Bethune, L., Busbridge, D., Grangier, D., Fini, E., El-Nouby, A., & Ablin, P. (2025). Scaling laws for optimal data mixtures. arXiv preprint arXiv:2507.09404.
>
> [3] Kaplan, J., McCandlish, S., Henighan, T., Brown, T. B., Chess, B., Child, R., ... & Amodei, D. (2020). Scaling laws for neural language models. arXiv preprint arXiv:2001.08361.
>
> [4] Chen, K., Gou, Y., Huang, R., Liu, Z., Tan, D., Xu, J., ... & Xu, H. (2025). Emova: Empowering language models to see,
> hear and speak with vivid emotions. In Proceedings of the Computer Vision and Pattern Recognition Conference (pp. 5455-5466).
>
> [5] Ye, J., Liu, P., Sun, T., Zhan, J., Zhou, Y., & Qiu, X. (2024). Data mixing laws: Optimizing data mixtures by predicting language modeling performance. arXiv preprint arXiv:2403.16952.

---

### Meta-Review · Area_Chair_2D3L · 2026-01-10

**Summary:**

Reviewer scores are 6, 2, 4, 6, and 4 (average 4.4). The main weaknesses include:
- limited novelty relative to prior data-mixing/scaling-law methods,
- technique issues, including high meta-cost to fit the regressor (1,620 runs) and unclear practical efficiency, and questions about generalizability across architectures and modalities
- limited experimental evaluation, including missing compute-cost reporting, lack of dynamic scheduling results despite advocating dynamic mixtures, and limited interpretation of the learned interaction parameters.

**Reviewer Concerns:**

The authors’ response adds related work, error bars, and compute/memory overheads for the regressor; clarifies novelty in moving to a cross-modal, compute-aware setting; provides qualitative interpretations of learned interaction terms and an ε-sensitivity ablation; and outlines how to produce phase-wise dynamic schedules from the same fitted law while arguing the exploration cost is a one-time amortized investment that extrapolates to 7B without refitting.

These revisions improve clarity and coverage, but they only partially resolve concerns about novelty, practicality of the meta-cost, breadth/strength of baselines, and the absence of dynamic-schedule results in the main experiments.

**Reviewer Scores:**

Reviewer wtc7 (6) and 3KNk (6) are likely to maintain their ratings.
Reviewer unVx may potentially slightly increase his rating from 4 to 6.
Reviewer ukzq (4) and f4MD (2) are likely to keep their ratings unchanged.

---

### Decision · Program_Chairs · 2026-01-26

Reject